# Selective activation of AKAP150/TRPV1 in ventrolateral periaqueductal gray GABAergic neurons facilitates conditioned place aversion in male mice

Xiaohui Bai[1,2,4], Kun Zhang[1,4], Chaopeng Ou [1,4], Bilin Nie[3,4], Jianxing Zhang[1], Yongtian Huang[1], Yingjun Zhang[1], Jingxiu Huang[1], Handong Ouyang [1✉], Minghui Cao [2✉] & Wan Huang [1✉]

Aversion refers to feelings of strong dislike or avoidance toward particular stimuli or situations. Aversion can be caused by pain stimuli and has a long-term negative impact on physical and mental health. Aversion can also be caused by drug abuse withdrawal, resulting in people with substance use disorder to relapse. However, the mechanisms underlying aversion remain unclear. The ventrolateral periaqueductal gray (vlPAG) is considered to play a key role in aversive behavior. Our study showed that inhibition of vlPAG GABAergic neurons significantly attenuated the conditioned place aversion (CPA) induced by hindpaw pain pinch or naloxone-precipitated morphine withdrawal. However, activating or inhibiting glutamatergic neurons, or activating GABAergic neurons cannot affect or alter CPA response. AKAP150 protein expression and phosphorylated TRPV1 (p-TRPV1) were significantly upregulated in these two CPA models. In AKAP150[flox/flox] mice and C57/B6J wild-type mice, cell-type-selective inhibition of AKAP150 in GABAergic neurons in the vlPAG attenuated aversion. However, downregulating AKAP150 in glutamatergic neurons did not attenuate aversion. Knockdown of AKAP150 in GABAergic neurons effectively reversed the p-TRPV1 upregulation in these two CPA models utilized in our study. Collectively, inhibition of the AKAP150/p-TRPV1 pathway in GABAergic neurons in the vlPAG may be considered a potential therapeutic target for the CPA response.

[1] Department of Anesthesiology, State Key Laboratory of Oncology in South China, Sun Yat-sen University Cancer Center, Collaborative Innovation Center for Cancer Medicine, Guangzhou, China. [2] Department of Anesthesiology, Guangdong Provincial Key Laboratory of Malignant Tumor Epigenetics and Gene Regulation. Sun Yat-sen Memorial Hospital, Sun Yat-sen University, Guangzhou, China. [3] Department of Anesthesiology, Guangdong Women and Children Hospital, Guangzhou, China. [4]These authors contributed equally: Xiaohui Bai, Kun Zhang, Chaopeng Ou, Bilin Nie. ✉email: ouyhd@sysucc.org.cn; caomh@mail.sysu.edu.cn; huangwan@sysucc.org.cn

The periaqueductal gray (PAG) is a highly conserved central gray region of the mesencephalon. It plays a key role in aversive behavior and integrates feelings of discomfort triggered by various reinforcers, such as pain, anxiety, fear and drug withdrawal[1–5]. The PAG comprises a diverse group of neuronal subpopulations that express a variety of neurotransmitters and neuropeptides, such as norepinephrine, dopamine, enkephalin, glutamate, mu-opioid receptors, and gamma-aminobutyric acid (GABA)[6]. Evidence indicates that subpopulations of the ventrolateral periaqueductal gray (vlPAG) exert different effects on nociception[7]. Activating GABAergic neurons or silencing glutamatergic neurons in the vlPAG can lead to allodynia, while silencing GABAergic neurons or activating glutamatergic neurons in the vlPAG can produce profound analgesia[7–9]. Activating dopamine neurons in the vlPAG not only produces profound analgesia but also produces fewer anxiety signs, indicating that vlPAG dopamine neurons are important targets for pain treatments[8]. The role of the vlPAG in CPA is not fully understood. PAG GABAergic neurons are common neurophysiological correlates of the negative valence system, and dysregulation of this population may contribute to some mental health disorders[10]. GABAergic neurons in the ventral PAG also enhance anxiety phenotypes and inhibit conditioned fear responses[10]. In a morphine withdrawal model, GABAergic neurons in the vlPAG showed hyperexcitability, and GABA neurotransmitter release increased without alteration in glutamate neurotransmitter release[11,12]. Pruritis-induced aversion has been found to be attenuated only by activating GABAergic neurons but not by silencing glutamatergic neurons in the vlPAG[13]. Recently, some reports also showed that BDNF-TrkB signaling and phosphorylated NR2B/ERK/CREB play a pivotal role in the pain-induced CPA model[14]. IFN-γ binds to its receptors on brain endothelial cells, induces CXCL10 expression and regulates inflammation-induced aversion[15]. These findings indicate that the development mechanisms of different CPA models are not the same, and the role of the vlPAG in CPA is complicated. Transient receptor potential vanilloid 1 (TRPV1), a calcium permeable ion channel, can work as a biomarker for neuropathic pain triggered by several chemicals and heat (>42 °C) and then integrate into nociceptive signals[16,17]. TRPV1 knockout mice showed that TRPV1 may take part in the development process of aversive behavior, such as facilitating conditioned or unconditioned fear and regulating long-term depression[18]. TRPV1 is expressed in several brain zones, including the nucleus accumbens[19], hippocampus[20] and ganglia[21]. In addition, TRPV1 has been demonstrated to be expressed in the PAG, and modulation of nociception transmission by TRPV1 in the PAG occurs in a subcolumn-specific manner[22–24]. Moreover, it has been shown that TRPV1 in the PAG is upregulated in murine models of cold stress-induced nociception and depression[25,26]. Activation of TRPV1 in the vlPAG can produce analgesia by facilitating the release of glutamate and activating rostroventromedial medulla OFF neurons[27].

A-kinase anchoring proteins (AKAPs), scaffolding proteins, have been reported to interact with key signaling enzymes toward selected substrates to exert vital biological functions[28,29]. Several AKAPs might be the mutual target of the same subcellular compartment; however, splice variants of the same AKAP gene can be differentially targeted[30–32]. Furthermore, certain phosphorylation events have been reported to be tightly controlled by AKAPs[33,34]. AKAPs can bind to the regulatory subunit of protein kinase A (PKA) and localize PKA to membrane sites of action, such as NMDA and glutamate receptor complexes[35]. Among them, AKAP150 has been demonstrated to mediate anchoring of PKA and protein kinase C (PKC) to TRPV1, which is required for phosphorylation of the channel[36,37]. The direct anchoring of PKA

to TRPV1 by AKAP150 may be critical in the development of thermal hyperalgesia[33]. In addition, a recent study suggested that antagonizing the AKAP79-TRPV1 interaction is useful for inhibiting inflammatory hyperalgesia[38]. To better elucidate the role of the vlPAG in CPA, we used two CPA models, hindpaw pinch pain-induced CPA and morphine withdrawal-induced CPA, to explore the role of different types of neurons in the vlPAG in CPA. Then, we explored the AKAP150/TRPV1 pathway in glutamatergic and GABAergic neurons of the vlPAG under CPA. Our study aimed to reveal the cell-type mechanism and to provide potential targets for attenuating aversion.

## Results

**AKAP150 is upregulated in the vlPAG in CPA models**. To elucidate the mechanisms of the aversion process, we used two aversion models in this study: hindpaw pinch pain CPA and naloxone-precipitated morphine withdrawal CPA (Fig. 1a–d). All the mice showed the same preference before establishment of the aversion model, and this baseline time was defined as the pre-paired time (t1) (Fig. 1e, i). After the aversion model was established, the mice spent significantly less time in the paired compartment. This was defined as the postpaired time (t2) (Fig. 1f, j). The aversion score (t2-t1) was also significantly reduced in the pinch pain CPA model (Fig. 1g) and morphine withdrawal CPA model (Fig. 1k). The movement of all mice did not significantly change before and after CPA (Fig. 1h, l). These data showed that both CPA models were successfully established. The IHC and western blotting results showed that AKAP150 expression in the vlPAG increased significantly in both the pinch pain CPA model and morphine withdrawal CPA model (Fig. 2a–d). This result indicates that AKAP150 in the vlPAG might be involved in the process of CPA.

**Inhibition of GABAergic neurons in the vlPAG attenuates the CPA response**. Our immunofluorescence assay showed that both glutamatergic neurons (CamkIIα marked) and GABAergic neurons (Vgat1 marked) exist in the vlPAG (Fig. 3a, b). To determine the functions of glutamatergic and GABAergic neurons during CPA, we used chemogenetic activation or inhibition in mice. We injected AAV of CaMKIIα-hM4D(Gi)-mCherry, CaMKIIα-hM3D(Gq)-mCherry, Vgat1-hM4D(Gi)-mCherry, and Vgat1-hM3D(Gq)-mCherry into the mouse vlPAG. We demonstrated that these viruses and the control viruses did not affect the basic preferences and movement ability of the mice (Supplementary Fig. 1a–d). Generally, AAV infection and stable expression take up to three weeks in mice. Thus, the mice were used to establish the pinch pain CPA model or morphine withdrawal CPA model 21 days after virus injection. There was no significant change in the aversion score when activating or inhibiting glutamatergic neurons in either CPA model (Fig. 3c, e). With manipulation of GABAergic neurons, the aversion score was obviously increased only in the Vgat1-hM4D(Gi) + CNO-treated group in the pinch pain CPA model (Fig. 3g) and morphine withdrawal CPA model (Fig. 3i) but not in the Vgat1-hM3D(Gq) + CNO-treated group (Fig. 3g, i). And the movement ability of all mice showed no significant differences before or after the CPA tests (Fig. 3d, f, h, j). These data show that inhibiting GABAergic neuron activation in the vlPAG is an effective method to attenuate the CPA response.

**Knockdown of AKAP150 in GABAergic neurons of the vlPAG attenuates aversion**. As AKAP150 in the vlPAG was upregulated in aversion models (Fig. 2a–d), we further conducted double immunofluorescence labeling to determine the specific cell location of AKAP150 in the vlPAG. Our immunofluorescence assay showed that AKAP150 colocalized with NeuN (a neuronal cell

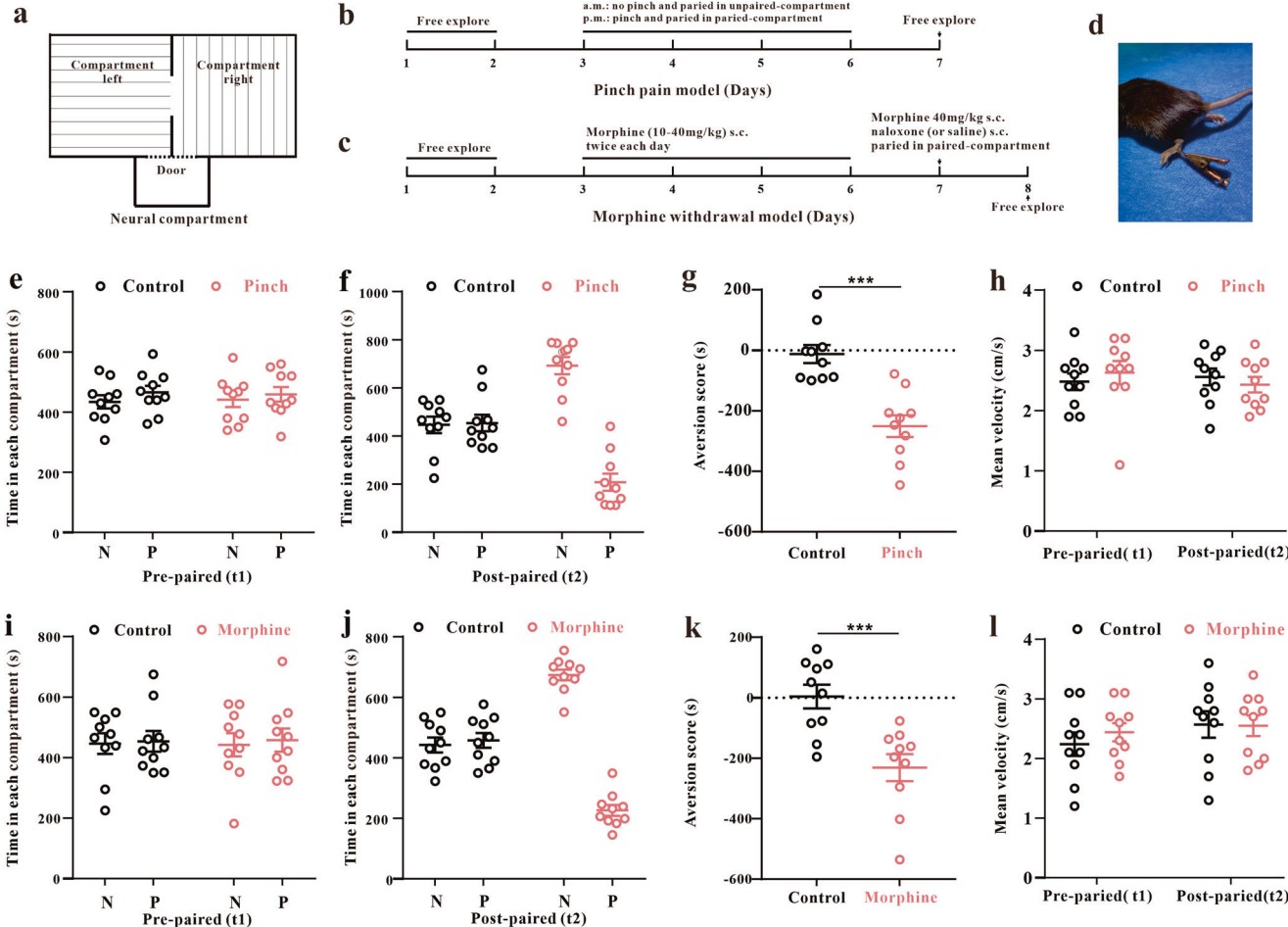

**Fig. 1 Establishment of the hindpaw pain pinch CPA model and naloxone-precipitated morphine withdrawal CPA model. a** Schematic of the three-compartment apparatus applied for the CPA test. Experimental timeline of the hindpaw pain pinch CPA model (**b**) and naloxone-precipitated morphine withdrawal CPA model (**c**) in mice. **d** Schematic of hindpaw pinch pain treatment in mice. **e–h** CPA test in mice with pinch pain. **e** Time spent in each compartment before pinch pain pairing (t1); **f** Time spent in each compartment after pinch pain pairing (t2); **g** The aversion score in the paired compartment (t2-t1), ***$P < 0.001$, the pinch pain group vs. the control group; two-sample $t$ test. **h** Mean velocity did not significantly change before and after pinch pain pairing ($F_{1, 18} = 0.662$, $P = 0.426$, two-way repeated-measures ANOVA, $n = 10$). **i–l** CPA test in mice with naloxone-precipitated morphine withdrawal. **i** Time spent in each compartment before morphine withdrawal pairing (t1); **j** Time spent in each compartment after morphine withdrawal pairing (t2); **k** The aversion score in the paired compartment (t2-t1), ***$P < 0.001$, the morphine withdrawal group vs. the control group; two-sample $t$ test. **l** Mean velocity did not significantly change before and after morphine withdrawal ($F_{1, 18} = 0.24$, $P = 0.63$, two-way repeated-measures ANOVA, $n = 10$). Data are shown as the mean ± SEM.

nuclei marker) and β-tubulin (a neuronal cell body marker) (Fig. 4a, b). Furthermore, we found that among the neurons in the vlPAG, both glutamatergic (CamkIIα marked) and GABAergic (Vgat1 marked) neurons expressed AKAP150 (Fig. 4c, d).

To test the functions of AKAP150 in the vlPAG in CPA, we downregulated AKAP150 in the vlPAG using CMV-AKAP150-shRNA, CamkIIα-AKAP150-shRNA, and Vgat1-AKAP150-shRNA in WT mice. CMV-Cre-EGFP, CamkIIα-Cre-EGFP, and Vgat1-Cre-EGFP were also used in AKAP150$^{fl/fl}$ mice to specifically knock down AKAP150. Twenty-one days later, the AAV transfection efficacy was stable, and all mice were subjected to the pinch pain CPA model and morphine withdrawal CPA model. We confirmed that these viruses did not affect the basic preference and movement ability of wild-type mice (Supplementary Fig. 2a, b) and AKAP150$^{fl/fl}$ mice for CPA (Supplementary Fig. 2c, d). As the CPA assay showed, compared with the CMV-shRNA-NC-treated group, the CMV-AKAP150-shRNA groups showed a higher aversion score in the pinch pain model (Fig. 4k) and morphine withdrawal model (Fig. 4m). In addition, compared with that of the

CMV-EGFP-treated group, the aversion score was obviously increased in AKAP150$^{fl/fl}$ mice after CMV-Cre-EGFP injection in the pinch pain model (Fig. 4o) and morphine withdrawal model (Fig. 4q). These data showed that knockdown of AKAP150 in the vlPAG can attenuate the CPA responses in both models. To further test the function of AKAP150 in glutamatergic and GABAergic neurons in the vlPAG under the CPA model, we knocked down AKAP150 in glutamatergic or GABAergic neurons in the vlPAG in a cell type-selective manner. In the present pilot, CamkIIα-AKAP150-shRNA and Vgat1-AKAP150-shRNA were injected into the WT mouse vlPAG, and AAV-CamkIIα-Cre-EGFP and AAV-Vgat1-Cre-EGFP were injected into the AKAP150$^{fl/fl}$ mouse vlPAG. After 21 days, when AAV transgenic expression was stable, the mice were subjected to the pinch pain CPA model or morphine withdrawal CPA model, and CPA tests were subsequently performed. The CPA assay indicated that knockdown of AKAP150 in GABAergic neurons in the vlPAG significantly inhibited the CPA response and increased the aversion score in the pinch pain model (Fig. 4k, o) and morphine withdrawal model (Fig. 4m, q). However, silencing AKAP150 in glutamatergic neurons did not

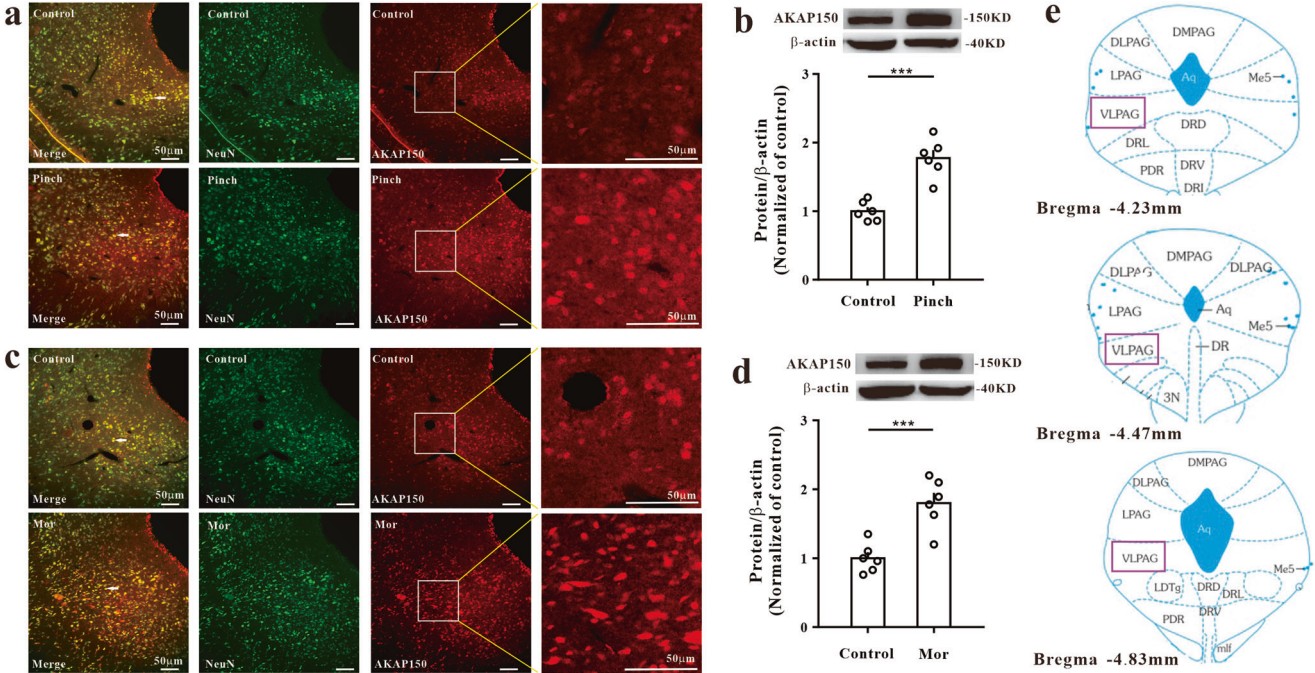

**Fig. 2 AKAP150 increased in the vlPAG in the CPA model.** Immunofluorescence images (**a**) and western blotting assay (**b**) showed that the expression of AKAP150 increased obviously in the pinch pain CPA model. Scale bar, 50 μm ($n = 3$). (***$P < 0.001$ vs. the control group, two-sample $t$ test, $n = 6$). Immunofluorescence images (**c**) and western blotting assay (**d**) showed that the expression of AKAP150 increased obviously in the morphine-withdrawal CPA model. Scale bar, 50 μm ($n = 3$). (***$P < 0.001$ vs. the control group, two-sample $t$ test, $n = 6$). **e** Schematic of the vlPAG in different layers of the mesencephalon. Data are shown as the mean ± SEM.

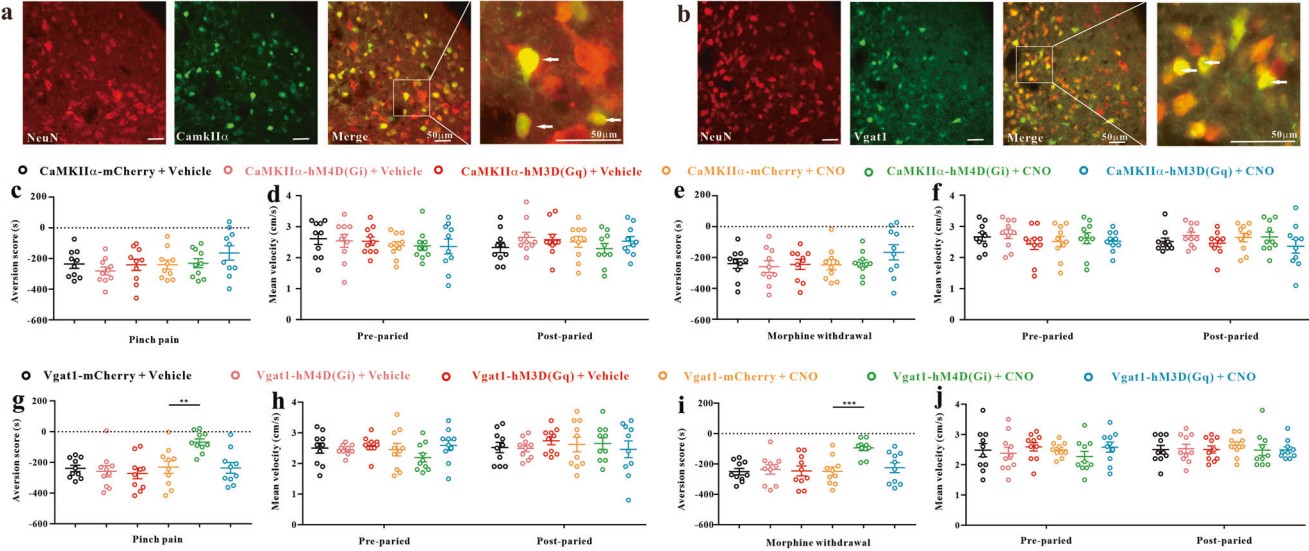

**Fig. 3 Chemogenetic manipulation of vlPAG neurons in the pinch pain CPA model and morphine withdrawal CPA model.** Immunofluorescence images showed that CamkIIα (**a**) and Vgat1 (**b**) were costained with NeuN, and the arrows indicated CamkIIα (**a**) and Vgat1 (**b**) were co-expressed with NeuN. Scale bar, 50 μm; ($n = 3$). **c–f** CPA tests in mice with inhibition or activation of vlPAG glutamatergic neurons. **c** Inhibition or activation of vlPAG glutamatergic neurons did not affect the aversion score in mice with pinch pain CPA ($F_{5, 54} = 1.282$ $P = 0.285$, one-way ANOVA, $n = 10$). **d** Mean velocity did not significantly change before and after the pinch pain CPA model ($F_{5, 54} = 0.440$, $P = 0.818$, two-way repeated-measures ANOVA, $n = 10$). **e** Inhibition or activation of vlPAG glutamatergic neurons did not affect the CPA score in mice with morphine withdrawal CPA ($F_{5, 54} = 0.871$, $P = 0.507$, one-way ANOVA, $n = 10$). **f** Mean velocity did not significantly change before and after morphine withdrawal in the CPA model ($F_{5, 54} = 0.319$, $P = 0.899$, two-way repeated-measures ANOVA, $n = 10$). **g–j** CPA tests in mice with inhibition or activation of vlPAG GABAergic neurons. **g** Inhibiting vlPAG GABAergic neurons, but not activating them, increased the CPA score in mice with pinch pain CPA ($F_{5, 54} = 5.380$, $P < 0.001$; ** $P < 0.01$ vs. AAV-Vgat1-mCherry + CNO group, one-way ANOVA, $n = 10$). **h** Mean velocity did not significantly change before and after pinched pain in the CPA model ($F_{5, 54} = 0.781$, $P = 0.568$, two-way repeated-measures ANOVA, $n = 10$). **i** Inhibiting vlPAG GABAergic neurons, but not activating them, increased the aversion score in mice in the morphine withdrawal CPA model ($F_{5, 54} = 4.751$, $P < 0.001$; ***$P < 0.001$ vs. AAV-Vgat1-mCherry + CNO group, one-way ANOVA, $n = 10$). **j** Mean velocity did not significantly change before and after morphine withdrawal in the CPA model ($F_{5, 54} = 0.459$, $P = 0.805$, two-way repeated-measures ANOVA, $n = 10$). Data are shown as the mean ± SEM.

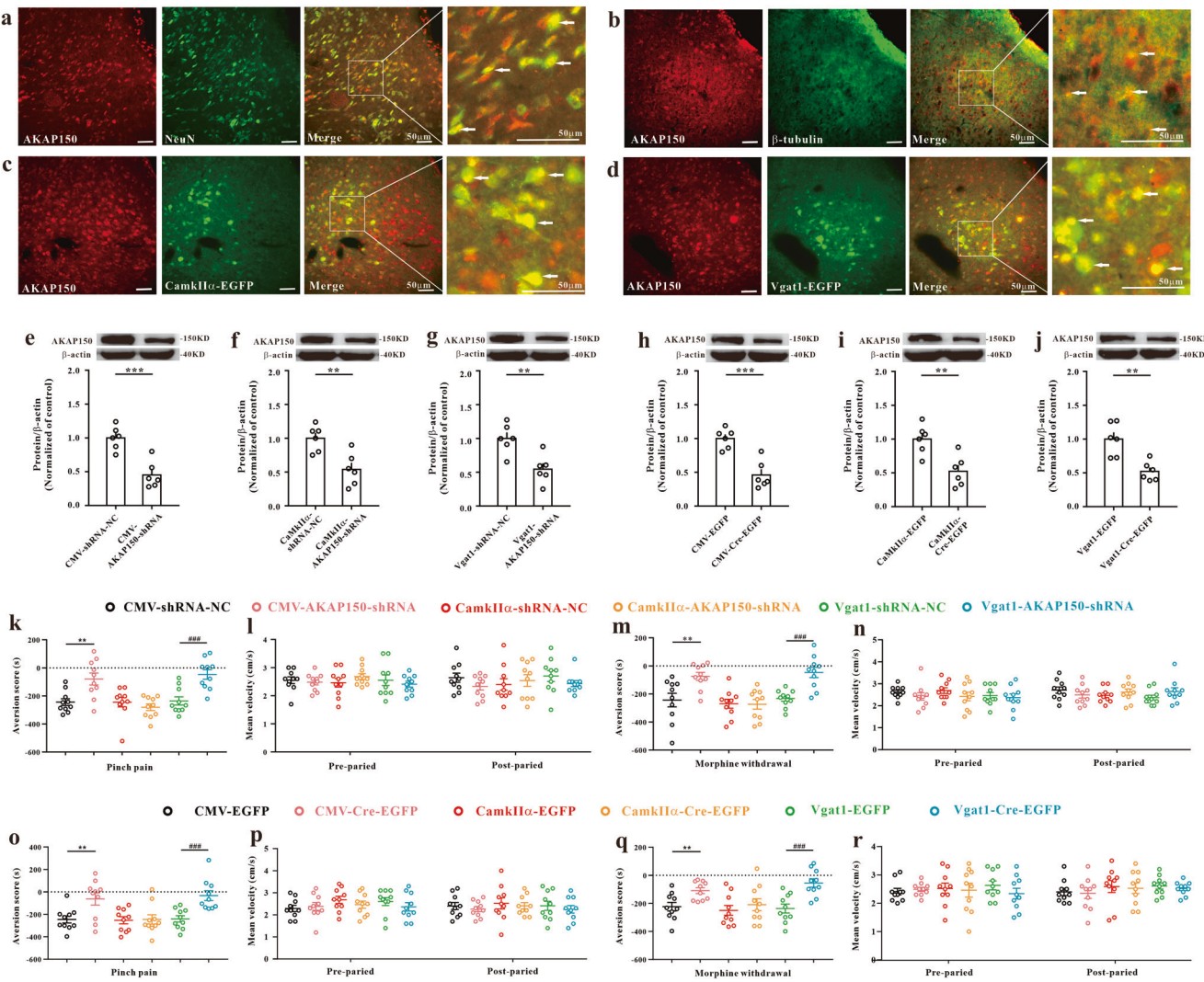

**Fig. 4 Knockdown of AKAP150 in the vlPAG in the pinch pain CPA model and morphine withdrawal CPA model.** Immunofluorescence images showed that AKAP150 was costained with NeuN (**a**), β-III tubulin (**b**), CamkIIα-EGFP (**c**), and Vgat1-EGFP (**d**), and the arrows indicated AKAP150 was co-expressed with NeuN (**a**), β-III tubulin (**b**), CamkIIα-EGFP (**c**), and Vgat1-EGFP (**d**). Scale bar, 50 μm, ($n = 3$). The expression of AKAP150 after CMV-AKAP150-shRNA (**e**), CamkIIα-AKAP150-shRNA (**f**), and Vgat1-AKAP150-shRNA (**g**) treatment in WT mice (**$P < 0.01$, ***$P < 0.001$ vs. the corresponding shRNA-NC group, two sample $t$ test, $n = 6$). The expression of AKAP150 after CMV-Cre-EGFP (**h**), CamkIIα-Cre-EGFP (**i**), Vgat1-Cre-EGFP (**j**) in AKAP150$^{fl/fl}$ mice. **k**–**n** CPA tests in WT mice with knockdown of AKAP150. **k** Global knockdown of AKAP150 or cell-type-selective knockdown of AKAP150 in GABAergic neurons, but not in glutamatergic neurons, significantly increased the aversion score in WT mice with pinch pain CPA ($F_{5, 54} = 9.589$, $P < 0.001$; **$P < 0.01$ compared with the CMV-shRNA-NC group, ###$P < 0.001$ compared with the Vgat1-shRNA-NC group, one-way ANOVA, $n = 10$). **l** No significant differences in mean velocity were observed in either group before or after CPA model pinch pain ($F_{5, 54} = 0.305$, $P = 0.908$, two-way repeated-measures ANOVA, $n = 10$). **m** Global knockdown of AKAP150 or cell-type-selective knockdown of AKAP150 in GABAergic neurons, but not in glutamatergic neurons, significantly increased the aversion score in WT mice with morphine withdrawal CPA ($F_{5, 54} = 8.836$, $P < 0.001$; **$P < 0.01$ compared with the CMV-shRNA-NC group; ###$P < 0.001$ compared with the Vgat1-shRNA-NC group, one-way ANOVA, $n = 10$). **n** No significant differences in mean velocity were observed in either group before or after morphine withdrawal in the CPA model ($F_{5, 54} = 0.734$, $P = 0.601$, two-way repeated-measures ANOVA, $n = 10$). **o**–**r** CPA tests in AKAP150$^{fl/fl}$ mice with knockdown of AKAP150. **o** Global knockdown of AKAP150 or cell-type-selective knockdown of AKAP150 in GABAergic neurons, but not in glutamatergic neurons, significantly increased the CPA score in AKAP150$^{fl/fl}$ mice with pinch pain CPA ($F_{5, 54} = 6.060$, $P < 0.001$; **$P < 0.01$ compared with the CMV-EGFP group; ###$P < 0.001$ compared with the Vgat1-EGFP group, one-way ANOVA, $n = 10$). **p** No significant differences in mean velocity were observed in either group before or after CPA model pinch pain ($F_{5, 54} = 0.196$, $P = 0.963$, two-way repeated-measures ANOVA, $n = 10$). **q** Global inhibition of AKAP150 or sequestration of AKAP150 in GABAergic neurons, but not glutamatergic neurons, significantly increased the CPA score in AKAP150$^{fl/fl}$ mice in the morphine withdrawal CPA model ($F_{5, 54} = 6.682$, $P < 0.001$; **$P < 0.01$ compared with the CMV-EGFP group; ###$P < 0.001$ compared with the Vgat1-EGFP group, one-way ANOVA, $n = 10$). **r** No significant differences in mean velocity were observed in either group before or after morphine withdrawal in the CPA model ($F_{5, 54} = 0.239$, $P = 0.944$, two-way repeated-measures ANOVA, $n = 10$). Data are shown as the mean ± SEM.

significantly affect the aversion score in the pinch pain CPA model (Fig. 4k, o) or morphine withdrawal model (Fig. 4m, q). And the movement ability of all mice showed no significant differences before or after the CPA tests (Fig. 4l, n, p, r). The AKAP150-shRNA target sequence and the inhibition effectiveness are shown in Supplementary Table 1 and Fig. 4e–j. These data suggest that AKAP150 expressed in GABAergic neurons in the vlPAG, but not glutamatergic neurons, regulates the CPA process.

**Silencing TRPV1 in GABAergic neurons of the vlPAG attenuates the CPA response.** A previous study suggested that AKAP150 plays an important role in mediating the phosphorylation of transient receptor potential channel type 1 (TRPV1)[39]. Therefore, we further measured and tested whether and how AKAP150-TRPV1 functions in CPA. Here, our immunofluorescence and western blotting assays showed that phosphorylated TRPV1 (p-TRPV1) increased significantly in pinch pain CPA model mice (Fig. 5c, d) and morphine withdrawal CPA model mice (Fig. 5e, f) but not total TRPV1 (Fig. 5a, b). Furthermore, we inhibited the expression of TRPV1 in GABAergic neurons of the vlPAG using Vgat1-TRPV1-shRNA. Compared with Vgat1-shRNA-NC, Vgat1-TRPV1-shRNA significantly attenuated the CPA response and increased the CPA score in the pinch pain model (Fig. 5h) and morphine withdrawal model (Fig. 5k) and inhibited the phosphorylated-TRPV1 protein level (Figs. 5i, j, 5l, m). Although the value of p-TRPV1/TRPV1 was not significantly different between Vgat1-shRNA-NC and Vgat1-TRPV1-shRNA, the p-TRPV1 protein level was verified to be downregulated through comparison with β-actin (Fig. 5j, m). The Vgat1-TRPV1-shRNA target sequence and its inhibitory effectiveness are shown in Supplementary Table 1 and Fig. 5g. We confirmed that the viruses did not affect the basic preference (Supplementary Fig. 2e, f), aversion score (Supplementary Fig. 2g), or movement ability of the mice (Supplementary Fig. 2h). These data suggest that silencing TRPV1 in GABAergic neurons of the vlPAG can attenuate the CPA response induced by pinch pain or morphine withdrawal.

**Phosphorylated TRPV1 expression is downregulated after silencing AKAP150 in GABAergic neurons of the vlPAG.** To further explore the role of AKAP150/TRPV1 in CPA, we measured the protein expression of pSer502-TRPV1 after inhibiting AKAP150. We applied Vgat1-AKAP150-shRNA in wild-type mice and Vgat1-Cre-EGFP in AKAP150[fl/fl] mice. Here, our immunofluorescence and western blotting data showed that pSer502-TRPV1 decreased significantly after silencing AKAP150 in GABAergic neurons of the vlPAG in WT mice (Fig. 6a–d) or AKAP150[fl/fl] mice (Fig. 6e–h). The upregulation of pSer502-TRPV1 in the vlPAG was greatly dependent on AKAP150 function, which was consistent with a previous study[39]. These data suggest that AKAP150 knockdown in GABAergic neurons of the vlPAG can reduce the expression of pSer502-TRPV1 and attenuate the aversion response.

## Discussion

The present study demonstrates that AKAP150 expression and p-TRPV1 are significantly increased in the pinch pain CPA model and morphine withdrawal CPA model. The GABAergic neurons, but not glutamatergic neurons, of the vlPAG function in the CPA response. Our study also demonstrates that silencing AKAP150

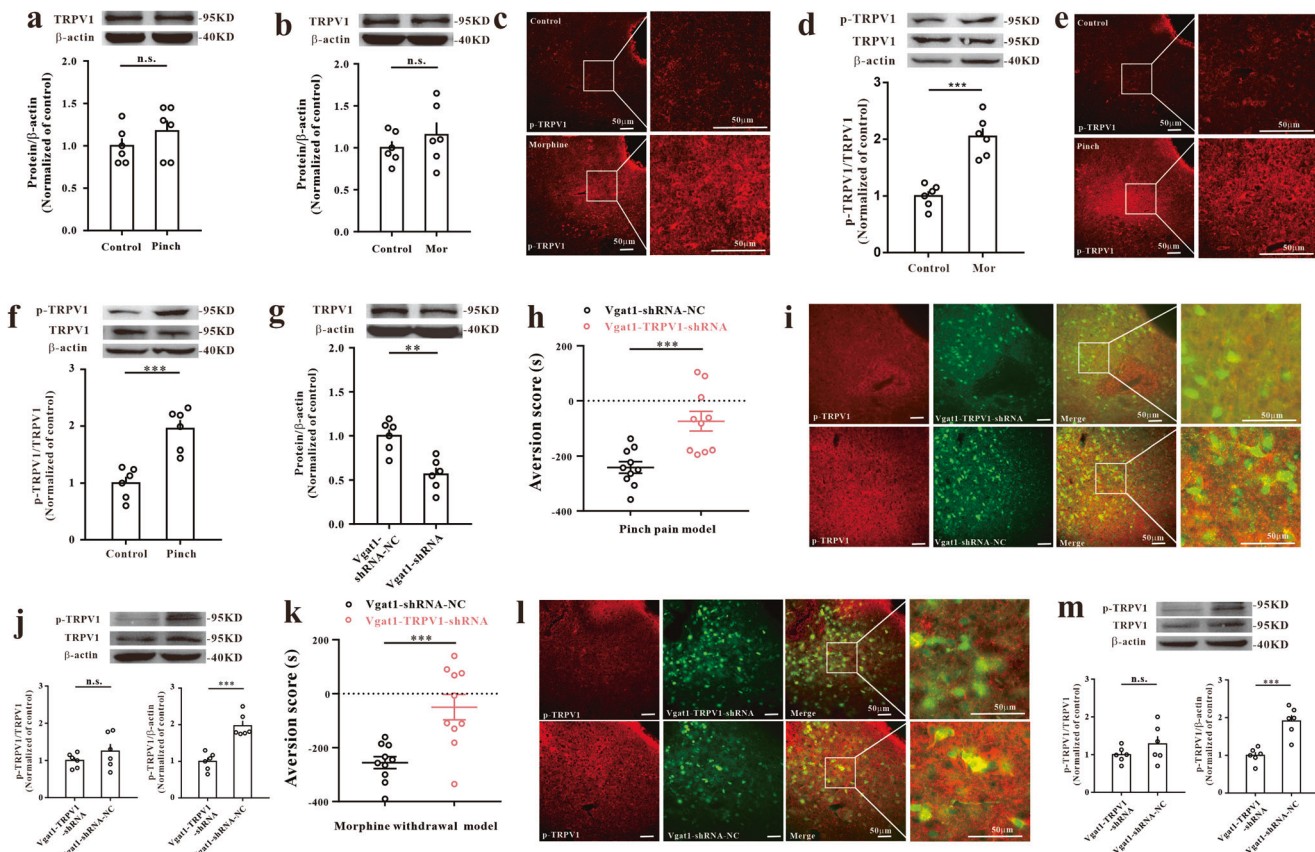

**Fig. 5 TRPV1 in GABAergic neurons of the vlPAG in CPA mice.** Expression of TRPV1 in the pinch pain CPA model (**a**) and morphine withdrawal CPA model (**b**). (two sample *t* test, *n* = 6). p-TRPV1 was upregulated in the pinch pain CPA model (**c**, **d**) and morphine withdrawal CPA model (**e**, **f**) [immunofluorescence image (**c**, **e**), scale bar, 50 μm (*n* = 3); western blotting (**d**, **f**), ***P < 0.001 vs. the control group, two sample *t* test, *n* = 6]. **g** Expression of TRPV1 after Vgat1-TRPV1-shRNA in WT mice (**P < 0.01 vs. Vgat1-shRNA-NC group, two sample *t* test, *n* = 6). Vgat1-TRPV1-shRNA treatment in mice [CPA tests for pinch pain CPA model (**h**) and morphine withdrawal model (**k**) mice, ***P < 0.001 vs. the Vgat1-shRNA-NC group, two sample *t* test, *n* = 10; immunofluorescence image (**i**, **l**), scale bar, 50 μm; *n* = 3; western blotting assay (**j**, **m**), n.s. No significance, ***P < 0.001 vs. the Vgat1-shRNA-NC group, two sample *t* test, *n* = 6]. Data are shown as the mean ± SEM.

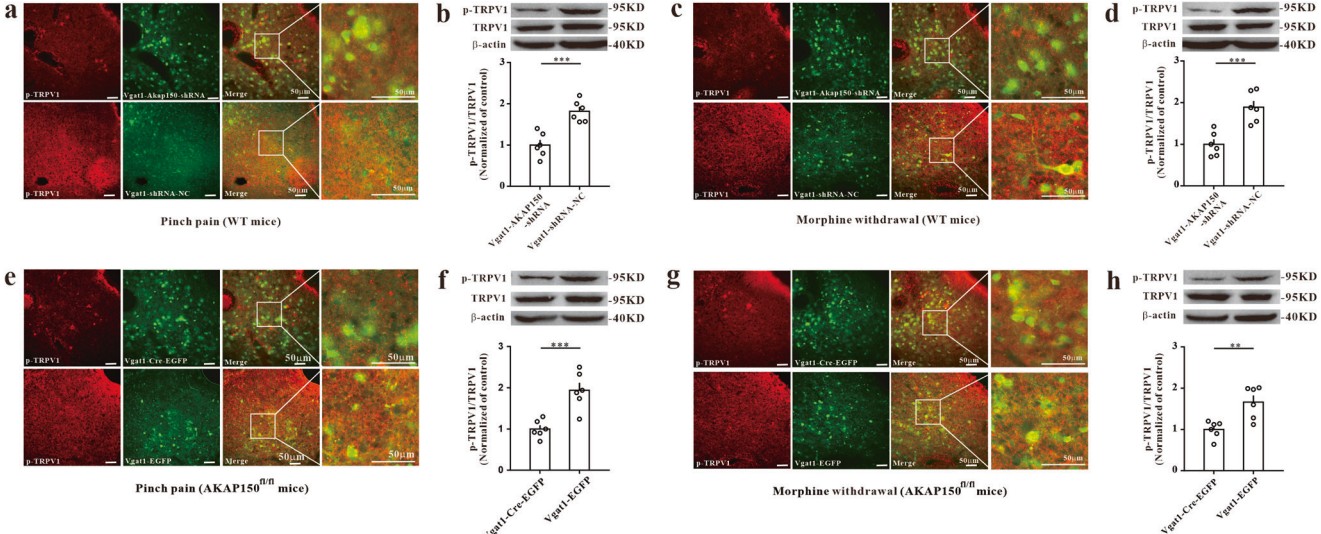

**Fig. 6 Downregulated AKAP150 resulted in the decreased phosphorylation of TRPV1 in the CPA models.** p-TRPV1 in the pinch pain CPA model (**a**, **b**) and morphine withdrawal model (**c**, **d**) with Vgat1-AKAP150-shRNA treatment in WT mice [immunofluorescence image (**a**, **c**), scale bar, 50 μm; $n = 3$; western blotting assay (**b**, **d**), **$P < 0.01$ vs. the Vgat1-shRNA-NC group, two sample $t$ test, $n = 6$]. p-TRPV1 in the pinch pain CPA model (**e**, **f**) and morphine withdrawal CPA model (**g**, **h**) with AAV-Vgat1-Cre-EGFP treatment in AKAP150$^{fl/fl}$ mice [immunofluorescence image (**e**, **g**). Scale bar, 50 μm; $n = 3$; western blotting assay (**f**, **h**), ***$P < 0.001$ vs. the vgat1-shRNA-NC group, two-sample $t$ test, $n = 6$]. Data are shown as the mean ± SEM.

or p-TRPV1 in GABAergic neurons of the vlPAG attenuates the pinch pain CPA and morphine withdrawal CPA response, and the upregulation of p-TRPV1 relies on AKAP150. We report that AKAP150/p-TRPV1 in GABAergic neurons of the vlPAG could serve as a potential target for pinch pain CPA and morphine withdrawal CPA.

Aversive stimuli such as pain and opioid withdrawal have a powerful impact on behavior and are considered to be the opposite valence of pleasure[40,41]. The stimuli associated with drug abuse, such as opioid abuse, might contribute to relapse after discontinuation of drug use[42]. Thus, it is necessary to determine the underlying mechanism of how conditions affect behaviors and to manage conditions therapeutically. Among various kinds of preclinical pain assessment tests, CPA is a well-known assay to assess motivational components of unpleasant stimuli and therapeutic effects[43]. To better elucidate the possible mechanisms of CPA development, we used two reliable models, the hindpaw pinch pain CPA model and the naloxone-precipitated morphine withdrawal CPA model.

The PAG is conserved across vertebrate species, and it is a vital structure involved in behaviors including feeding behavior, nociception modulation, autonomic regulation, visceral functions and aversion response[13,44,45]. The vlPAG in rodents coordinates passive coping behaviors and opioid analgesia. Evidence has demonstrated that glutamatergic and GABAergic neurons in the vlPAG have different effects on some physiological processes[7,46]. Inhibition of vlPAG GABAergic neurons or activation of glutamatergic neurons results in antinociception, whereas activation of vlPAG GABAergic neurons or inhibition of glutamatergic neurons leads to hypersensitivity to painful stimuli[7]. In our study, we demonstrate that activating glutamatergic neurons, inhibiting glutamatergic neurons or activating GABAergic neurons cannot affect or alter the CPA response. The results that inhibition of GABAergic neurons attenuates aversion while activation of GABAergic neurons does not affect aversion are considered reasonable, as inhibition and activation of the same neurons are not always exerting opposing effects. Acute pain stimulus is not necessarily equivalence to aversion. We assume that activation of GABAergic or glutamatergic neurons or

inhibition of glutamatergic neurons interferes with acute pain modulation but not aversion induced by hindpaw pain or morphine withdrawal. The activity of glutamatergic and GABAergic neurons in the PAG modulates itch in an opposing manner[13]. In our study, activation of GABAergic neurons, activation of glutamatergic neurons or inhibition of glutamatergic neurons did not affect the CPA response. Samineni[13] reported that pruritis-induced aversion is attenuated by activating GABAergic neurons but not by silencing glutamatergic neurons in the vlPAG. There are some discrepancies with our results. Our CPA assay reported that inhibition of GABAergic neurons attenuated the aversion response in the pinch pain CPA model and morphine withdrawal CPA model, while activating GABAergic neurons attenuated the aversion response in Samineni's study. Itch and nociception exhibit an inverse and reciprocal relationship at the behavioral level[13]. Although itch and nociception are processed and transmitted by similar neuroanatomical substrates at peripheral, spinal and supraspinal sites, they are discriminated and processed distinctly at the cellular level[13]. This may be the possible explanation for the significant difference between the two results.

According to our previous study, we found that AKAP150 has an important function in the development of neuropathic pain by regulating calcineurin and IL-10 expression[28]. However, whether and how AKAP150 in the vlPAG is involved in CPA are still unknown. In this study, the CPA assay clarified that AKAP150 in glutamatergic neurons in the vlPAG did not ameliorate the aversion response, while inhibiting AKAP150 in GABAergic neurons greatly attenuated aversion. This phenomenon indicates that AKAP150 in different neurons in the vlPAG exerts different functions during the CPA process.

AKAP150 is closely related to the function of TRPV1 and regulates TRPV1 phosphorylation by orienting PKA, PKC and phosphatase 2B toward membrane-associated substrates[47,48]. TRPV1 is also involved in calcium signaling fundamental for various cellular processes[49] and nociception regulation[36,50]. Ablation of TRPV1 terminals did not affect withdrawal responses to von Frey filament and noxious cold stimulations but caused a reduction in licking evoked by skin pinching, cold or hot plate stimulation, or skin burn injury[51]. This may be the theoretical

basis for TRPV1 regulating pinch pain CPA. TRPV1 is also found in various brain areas, including dopaminergic neurons of the substantia nigra, the PAG, hippocampal pyramidal neurons, hypothalamic neurons, the locus coeruleus in the brainstem and various layers of the cortex, where it might be involved in the modulation of synaptic plasticity[52,53]. Blocking TRPV1 suppresses morphine-induced c-fos expression, reduces opioid tolerance and opioid-induced hyperalgesia, attenuates opiate-mediated conditioned place preference (CPP) behaviors, decreases anxiety-like behaviors after abstinence from morphine, alleviates negative emotions during withdrawal, and suppresses the motivational properties of morphine and its reinstatement or relapse[54–57]. We observed that phosphorylated TRPV1 in the vlPAG was dramatically upregulated in the mice with CPA compared to the control mice. We downregulated TRPV1 in GABAergic neurons with Vgat1-TRPV1-shRNA and found that this intervention increased the CPA score. The results indicate that targeting TRPV1 phosphorylation in GABAergic neurons is an effective method to attenuate the CPA response in mice. Liao's[22] results showed that activating TRPV1 on glutamatergic neurons in the vlPAG enhances mGluR5-mediated 2-arachidonolyglycerol expression and then causes the retrograde disinhibition of GABAergic neurons, resulting in activation of the descending inhibitory nociception pathway and antinociception. However, we did not evaluate the effect of TRPV1 overexpression on glutamatergic neurons and GABAergic neurons. We only tested the protein expression of phosphorylated TRPV1 after the inhibition of AKAP150 in GABAergic neurons in the vlPAG of CPA mice. Our results showed that TRPV1 phosphorylation levels are decreased by AKAP150 inhibition. Starowicz et al.[27] reported that a TRPV1 antagonist in the vlPAG facilitates nociceptive responses, while a TRPV1 agonist in the vlPAG promotes antinociception. This seems to be inconsistent with our study, since we state that only silencing TRPV1 in GABAergic neurons of the vlPAG can attenuate the CPA response induced by pinch pain or morphine withdrawal. However, there are differences between these two studies in experimental design. First, the aversion response induced by pain is not equivalent to the pain stimulus, meaning that the aversion and pain stimuli may both relate to the activity of the vlPAG but may not receive modulation from the same region in the vlPAG. Second, Starowicz's study[27] mainly investigated the instant effect of nociception or antinociception after injecting the TRPV1 antagonist or agonist, while our study selectively knocked down the expression of TRPV1 by virus overexpression with the GABA neuron promoter Vgat1. Neither TRPV1 agonist nor antagonist is selective toward TRPV1 receptors, which is completely different from the selective knockdown by shRNA. There are possibilities that TRPV1 in the vlPAG mediates the CPA response through other pathways, which requires further experiments for clarification.

In addition to TRPV1 phosphorylation, we propose several possible mechanisms by which AKAP150 regulates the CPA process. One possible mechanism is that AKAP150 may function through NMDA and AMPA receptors, as a previous study indicated that AKAP150 can affect synaptic function by NMDAR-dependent long-term potentiation (LTP) and long-term depression (LTD)[58]. Some research has shown that the aversion score can be blocked by microinjection of an NMDA receptor antagonist into the ventral tegmental area (VTA)[59]. Another possible mechanism is that AKAP150 may mediate aversion not only in the vlPAG but also in other brain regions, such as the basolateral amygdala and VTA[60,61]. Third, we only explored the function of AKAP150 in GABAergic and glutamatergic neurons in the vlPAG in aversion. It is also possible that AKAP150 mediates the CPA response by other cells, such as dopamine neurons, as a previous study indicated that dopamine neurons are

activated in both acute morphine withdrawal and withdrawal memory retrieval[61].

There are some limitations in the present study. First, several phosphorylation sites on TRPV1 have been identified, and we only investigated the Ser502 site. We are not sure whether phosphorylation sites other than Ser502 participate in the process by which pTRPV1 mediates CPA. Second, we did not provide physiological readout of vlPAG activity because it is beyond our technical capabilities. Third, we simply verified that the level of TRPV1 phosphorylation was altered after knockdown of AKAP150 expression, and we could not reveal the detailed and specific interaction between AKAP150 and TRPV1. Fourth, we did not include the patch clamp in the experimental design; thus, it is difficult for us to determine whether the impact on the CPA response is due to altered glutamatergic and GABAergic neuron activity or the effects on synaptic terminal release of neurotransmitters. Fifth, we have only explored the effect of AKP150 and pSer502-TRPV1 on the CPA response and have not excluded the possibility that AKAP150 might affect CPA through other mechanisms, such as calcineurin, PKA, and PKC. Further confirmational studies are needed.

Our present study showed that the vlPAG plays an important role in the process of hindpaw pinch pain-induced CPA and naloxone-precipitated morphine withdrawal-induced CPA. Inhibition of the AKAP150/p-TRPV1 pathway in GABAergic neurons in the vlPAG may effectively attenuate the CPA response and may be considered a potential therapeutic target for the CPA response.

## Methods

**Animals**. Male C57/BL6 mice (8–10 weeks) were obtained from the Institute of Guangdong Medicine Experimental Animal Center. AKAP150$^{flox/flox}$ mice were purchased from the Jackson Laboratory and were verified in our previous research[28]. Every five mice were kept in a cage with the humidity maintained at 50–60% and the temperature maintained at 25 ± 1°C. All animals were provided free access to water and food. All experimental protocols were approved by the Institutional Animal Care and Use Committee of Sun Yat-sen University Cancer Center (No. L102012020000X). The researchers were blinded to the animal behavior tests after drug treatment.

**Conditioned place aversion (CPA)**. The aversion score was measured using the CPA paradigm, which consisted of three compartments: two large conditioning compartments (15 cm × 15 cm × 30 cm) and a small neural compartment (10 cm × 5 cm × 10 cm) (Fig. 1a). In the left conditioning compartment, the walls were decorated with black−white transverse striations, and the floor was smooth. In the right conditioning compartment, the walls were decorated with black−white vertical striations, and the floor was frosted. The mice were allowed to freely enter any large compartment from the neural compartment. When the mice entered the large compartment, the door was closed quietly and automatically. Then, the mouse's movements were recorded and analyzed for 15 min. To balance the CPA assay and avoid the possible effect between the left and right compartments, we randomly assigned any one conditioning compartment as a paired compartment for each mouse. To display the data more conveniently and scientifically, we defined all the unpaired compartments as the "N-paired compartment" and all the paired compartments as the "paired compartment". The mice were given a hindpaw pinch or naloxone subcutaneous (s.c.) injection (Fig. 1b, c) in the paired compartment. The analysis software was used to record and analyze the time and moving path in the conditioning compartments (JLBehv-CPPM-4, Jiliang Technology Co., Ltd. Shanghai, China). The CPA score was defined as the time (postpaired) spent in the paired compartment minus the time (prepaired) spent in the paired compartment. Animals with a strong initial preference for either compartment (one compartment >720 s) were eliminated from the study (approximately 2% of all animals).

**Hindpaw pinch pain induced CPA**. The hindpaw pinch pain-induced CPA was conducted as followed[51]. An alligator clip was applied to the ventral skin surface between the footpad and the heel (Fig. 1d). Days 1 and 2 were set as baseline days, and each mouse could freely explore both compartments for 15 minutes. Days 3–6 were the paired days, and each mouse was confined to the N-paired compartment and grabbed three times at 5 min intervals without pinch stimulation in the morning (8 a.m.–11 a.m.). The mouse was confined to the paired compartment and received three trials of hindpaw ventral skin pinch: 1 min for each trial with a 4 min

interval (3 p.m.–5 p.m.). The mice were allowed to freely explore on Day 7 to evaluate the CPA score.

**Naloxone-precipitated morphine withdrawal-induced CPA**. The morphine withdrawal CPA was conducted as followed[62,63]. The conditioning procedure comprised a preconditioning session, four consecutive drug treatment days, a conditioning day and a posttest day. On Days 1 and 2, each mouse was placed in the neural compartment and allowed to explore the two large compartments with no restrictions. On Days 3 and 6, all mice were randomly divided into two groups. The morphine group was injected (s.c.) with morphine in escalating drug doses twice per day (8 a.m. and 4 p.m.), 10 mg/kg on the 1st day; 20 mg/kg on the 2nd day; 30 mg/kg on the 3rd day; and 40 mg/kg on the 4th day. Then, the mice were returned to their home cages. On the morning of Day 7, the mice were injected with 40 mg/kg morphine (s.c.), and one hour later, naloxone (1 mg/kg, s.c.) was added, and the mice were immediately confined to the paired compartment for 1 hour. The control group was also given the same dose of morphine, but saline (1 mg/kg, s.c.) was used to replace naloxone. On Day 8, the mice were kept in the CPA apparatus and allowed to explore freely, and the CPA score was recorded and analyzed (Fig. 1c).

**Surgeries and stereotaxic injections**. Under continuous isoflurane inhalation anesthesia, mice were placed in a stereotaxic frame (RWD Life Science Co., Ltd.), and body temperature was maintained at 36 °C with a heating pad. After extra local analgesia with a 1 ml subcutaneous injection of 1% lidocaine above the skull, the skull was fully exposed and perforated with a stereotaxic drill at the desired coordinates relative to bregma. Viral solution (250 nl) was infused using a microinjector with a 33 G needle for 10 min. After infusion, the needle was kept at the injection site for more than 15 min before being slowly withdrawn. For viral injections and implants, the following stereotaxic coordinates were used: vlPAG: AP: −4.45; ML: ± 0.55; DV: −2.7 taken relative to the dura (Fig. 2e)[64]. The test part of the PAG in the experiment was the caudal vlPAG part.

**Chemogenetic activation and inhibition**. For chemogenetic activation and inhibition, mice were injected with rAAV-CaMKIIα-hM4D(Gi)-mCherry, rAAV-CaMKIIα-hM3D(Gq)-mCherry, rAAV-Vgat1-hM4D(Gi)-mCherry, rAAV-Vgat1-hM3D(Gq)-mCherry and the corresponding control virus (Supplementary Table 2) in the vlPAG. The GABA neuron promoter we used was Vgat1, which is generally used to identify GABA interneurons but not projection neurons in the vlPAG. Twenty-one days later, when the virus expression was stable, all the mice were allowed to freely explore both sides of the CPA training apparatus for 15 min to evaluate their baseline place aversion. Thirty minutes before the CPA test, the mice received CNO (ApexBio) (3 mg/kg, i.p.) to manipulate vlPAG neuron activation or inhibition. CNO was dissolved in DMSO and diluted to the appropriate ratio with physiological saline. An equal volume of 0.05% DMSO was used as a vehicle control. The pinch pain CPA or morphine withdrawal CPA models were established as mentioned above. Briefly, mice were injected with CNO or DMSO on Days 3–6 under the pinch pain model. The mice received CNO or DMSO injection on Day 7 under the naloxone-precipitated morphine withdrawal CPA model (Fig. 1b, c).

**AKAP150 and TRPV1 knockdown**. We used AAV-shRNA virus to stably conditionally knock down the expression of AKAP150 or TRPV1 in the wild-type mouse vlPAG. We stereotaxically injected rAAV-CMV-AKAP150-shRNA-mCherry, rAAV-CaMKIIα-AKAP150-shRNA-mCherry, rAAV-Vgat1-AKAP150-shRNA-mCherry, rAAV-Vgat1-TRPV1-shRNA-mCherry and the corresponding control virus (Supplementary Table 2) into the vlPAG to conditionally knock down the expression of AKAP150 or TRPV1 in wild-type mice. We also stereotaxically injected rAAV-CMV-Cre-EGFP, rAAV-CaMKIIα-Cre-EGFP, rAAV-Vgat1-Cre-EGFP and the corresponding control virus (Supplementary Table 2) into the vlPAG to conditionally knock down the expression of AKAP150 in AKAP150[fl/fl] mice. The knockdown effectiveness in the vlPAG was verified using western blotting (Figs. 4e–j, 5g, j, m). AAV infection and stable gene expression take up to 3 weeks, and the baseline CPP determination of all mice is performed between 21 and 22 days after AAV injection.

**Immunohistochemical analyses**. Mice were perfused with 4% paraformaldehyde under anesthesia. The brain tissues were cut into 20 μm-thick sections after 30% DEPC-sucrose dehydration at 4 °C. Primary antibodies against AKAP150 (Santa Cruz, Cat# sc-377055, 1:200), phospho-TRPV1 (Ser502) (Affinity Biosciences, Cat# AF8520, 1:200), β-III Tubulin antibody (Abcam, Cat# ab78078, 0.4 μg ml⁻¹) and NeuN (Millipore, Cat# MAB377, 1:200) diluted in hybridization solution were then incubated with the vlPAG sections at 4 °C overnight; the sections were subsequently incubated with fluorescein secondary antibodies with Cy3 or Alexa 488 at room temperature (approximately 26 °C) for 1 h. Finally, the sections were stained with DAPI and imaged using a confocal microscope (Nikon) equipped with a digital camera.

**Western blotting**. The proteins extracted from the vlPAG were quantified using the BCA Protein Assay kit, separated by SDS-PAGE, and transferred onto PVDF membranes. TBST with 3% skim milk was applied to block the PVDF membranes with gentle shaking at room temperature (approximately 26 °C) for 1 h to avoid nonspecific binding. Primary antibodies against β-actin (Proteintech, Cat# 66009-1, 1:5000), AKAP150 (Santa Cruz, Cat# 377055, 1:200), TRPV1 (Affinity Biosciences, Cat# DF10320, 1:500) and phospho-TRPV1 (Ser502) (Affinity Biosciences, Cat# AF8520, 1:200) were then incubated with the PVDF membranes at 4 °C overnight. After the membrane was rinsed 3 times with TBST, the membrane was incubated for 1 h with anti-rabbit IgG secondary antibody (Abcam, ab6721, 1:10,000) at room temperature. Immunostained bands were detected using Immobilon Western Chemiluminescent HRP Substrate (Millipore, WBKLS0500) and Image Quant LAS 4000 mini system (GE Healthcare). Band intensities were analyzed using ImageJ and normalized to those of GAPDH or β-actin. All original western blots are shown in the Supplemental Materials (Supplementary Fig. 3). We equally divided each sample into two to run the blots of the loading controls and the proteins of interest. The loading controls were run on separate blots from the proteins of interest. Two hours after the last match of the pinch pain model (Fig. 1b Day 6) or morphine withdrawal model (Fig. 1c Day 7), the mice were anesthetized and euthanized for WB and IHC vlPAG specimen harvest.

**Statistics and reproducibility**. All data are shown as the mean ± standard error of the mean (SEM). SPSS software (version 25.0) was used to analyze the data. The comparison of behavior and WB between two groups was conducted using t tests. The comparison of mean velocity was conducted using two-way repeated-measures ANOVA followed by Bonferroni's post hoc test. The comparison of aversion scores among multiple groups was conducted using one-way ANOVA followed by Bonferroni's post hoc test. The criterion for statistical significance was set at $P < 0.05$.

**Reporting summary**. Further information on research design is available in the Nature Portfolio Reporting Summary linked to this article.

## Data availability

All data supporting the findings of this study are available within the paper and its Supplementary Information. The numerical source data for the graphs in the manuscript are available in the Supplementary Data file.

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

## Acknowledgements

This work was supported by the National Natural Science Foundation of China (82071237 and 81771192), the Natural Science Foundation of Guangdong Province (2023A1515010147 and 2023A1515011180) and OUYHD's Clinical Medical Scientists Project of Sun Yat-sen University Cancer Center.

## Author contributions

W.H., X.B. and H.O. designed the experiment. C.O., X.B. and H.O. wrote the manuscript. W.H., K.Z., B.N., J.Z., Y.H., Y.Z. and J.H. performed the experiments and collected the data. X.B., H.O. and K.Z. performed the analysis. M.C. and H.O. supervised this study.

## Competing interests

The authors declare no competing interests.
