## [Peer Review File · Communications Biology]

Reviewers' comments:

Reviewer #1 (Remarks to the Author):

In manuscript COMMSBIO-23-0169-T, entitled "AKAP150/TRPV1 in GABAergic neurons rather than glutamatergic neurons in the PAG is involved in conditioned place aversion in mice", the authors contend that AKAP modulates TRPV1 phosphorylation in GABA neurons of the vIPAG to control conditioned place aversion. In this study, the authors utilize a well-documented model of rodent aversion to pinch and morphine withdrawal. Furthermore, they employ viral genetics to control AKAP and TRPV1 expression in PAG neurons via stereoscopic injection of AKAP-floxed mice. The experiments presented here provide some sound support of the authors' hypotheses, while other results fail to utilize correct controls to produce any conclusive evidence. Furthermore, there is an apparent lack of synonymy between the text of the paper, figure legends, and actual figures that has the potential to exponentially increase reader ambiguity, making the entire paper difficult to follow. The following are more specific points that bear additional examination:

1. The investigation of TRPV1 phosphorylation is mishandled. Firstly, there is no rationale as to why Ser502 is investigated, is there a specific kinase that AKAP is scaffolding that should control phosphorylation at that site? Furthermore, the WB analyses of p-TRPV1 is incorrect (Fig 4 and 5), as a protein's phosphorylation status should always be normalized to total amounts of said protein, not beta-actin. This would ensure that changes in reported phosphorylation are not due to changes in the amount of protein being phosphorylated.
2. The main point of this submission, and the name of the title, implies that AKAP/TRPV1 together modulate the behavior phenomenon, while the data only implies that they both do so, but separately. There is no experiment performed in this project to underscore any premise that AKAP is scaffolding anything to TRPV12 to control the behavior. There is only a collection of individual studies that show either AKAP or TRPV1 have modulatory roles, not that these roles are in any way connected.
3. In Fig 1, the authors require an IHC control for panels N and P. Perhaps NeuN as used elsewhere in the submission.
4. Controls for AKAP shRNA efficacy should be alluded to earlier in the results section corresponding to Fig 3. The authors invite reader ambiguity by not plainly identifying supporting data that is necessary to validate their animal model.
5. In the results section corresponding to Fig 5, the authors state that AKAP knockdown can reduce the expression of p-TRPV1. Do the authors mean for pSer502? pSer800? Perhaps pSer116? Maybe one of the phosphorylatable Tyr residues of TRPV1. This invites more ambiguity by the reader to determine what the authors conclude.

Reviewer #2 (Remarks to the Author):

In this report the authors study the involvement of TRPV1 receptors, the anchoring protein AKAP150, and GABAergic versus glutamatergic inputs to the ventrolateral periaqueductal gray (vIPAG) in 2 distinct conditioned place aversion (CPA) models. The first model examines CPA caused by hind paw pinch, and the second by naloxone-precipitated opiate withdrawal. The rationale for examining AKAP150 is derived from observations that it can interact with signaling enzymes such as the regulatory subunit of protein kinase A (PKA) and PKC to anchor these enzymes and phosphorylate

TRPV1Rs, which are ubiquitously involved in nociception. Using a variety of biochemical and genetic tools, including chemogenetics, and protein knockdown, the authors report that selective inhibition of vIPAG GABAergic neurons reduced CPA in both models, whereas inhibition of vIPAG glutamatergic neurons had no effect on the pain or morphine withdrawal CPA responses. The authors also show that AKAP150 protein expression levels and phosphorylation of TRPV1 were increased in both CPA models, and that selective knockdown of AKAP150 with shRNA in GABAergic neurons also decreased both forms of CPA. In general, this is a thorough and rigorous study that provides evidence for the involvement of AKAP150 in the phosphorylation of TRPV1 channels in GABAergic neurons of the vIPAG. This suggests that this pathway may be a candidate for consideration as a target for alleviating conditioned aversion.

Comments.

1. It is difficult to determine from the data or the discussion whether the authors believe that it is the synaptic input to vIPAG neurons that is important for determining CPA, or whether it is the effect of their manipulations on the glutamate or GABA neurons themselves within the vIPAG that is important. Can the chemogenetic manipulations discriminate altered glutamatergic and GABAergic neuron activity versus effects on synaptic terminal release of these neurotransmitters? Specifically, can the authors state whether these manipulations targeted neurotransmitter release in the vIPAG, or just the somatic activity of GABAergic versus glutamatergic neurons in the vIPAG? As there is no direct physiological readout of the activity of these neurons or of synaptic transmission, it would be useful to either provide this, or to address this issue in some other way, such as commenting on it in the discussion.
2. Figures 2-3 are unnecessarily complex because they show all of the pre-paired and post-paired data, rather than just providing a single example of this followed by the simplified aversion score. I think that it is sufficient to show the pre- and post-paired data once as an example, and then provide the aversion score data in the rest of the figures.
3. line 66. "this finding indicated that different CPA model mechanisms are not the same, and the role of PAG in CPA is complex and complicated." As the terms "complex" and "complicated" mean the same thing in this context, this sentence should be rewritten for clarity.
4. line 250. "after 21 days, when AAV transgenic expression was stable,". Do the authors mean to say, "when the expression of gene expression was stable"? This should be corrected.

Reviewer #3 (Remarks to the Author):

The present study examined the role of AKAP150/TRPV1 pathway in glutamatergic and GABAergic neurons in the ventrolateral periaqueductal gray (vIPAG) during pinch pain and morphine withdrawal CPA models. The authors report that the expression of AKAP150 and p-TRPV1 in the vIPAG was increased after CPA testing paired with paw pinch or morphine withdrawal. Using chemogenetics, they showed that inhibition of GABA (but not glutamate) neurons attenuated both types of CPA, and that AKAP150 knockdown in GABA (but not glutamate) vIPAG neurons attenuated both types of CPA and p-TRPV1 expression, and that TRPV1 knockdown in vIPAG GABA neurons also attenuated both types of CPA in mice. Overall, these findings suggest that AKAP150 and TRPV1 in vIPAG GABA neurons are important for mediating the aversive components of pain and morphine withdrawal. In general, this is a parametrically complete set of studies that are well controlled, and that report a very interesting and

informative set of results regarding cells and circuits underlying aversive aspects of opioid withdrawal and acute pain. However, there are many issues that need to be addressed. The manuscript requires a substantial re-write, although there is no need for new experiments.

1. In the title:

a. State "in male mice."

b. "is involved in" is not informative... state direction of the result.

2. The manuscript is extremely hard to follow, and there many unclear points and details. Please carefully re-write, perhaps with the assistance of a native English speaker. For example, in the Abstract and in the manuscript more generally, there are some sweeping statements and definitions that are inaccurate, and also loose swapping of terminology, lack of clarity, etc., for example:

a. The Abstract defines aversion as "the process by which a noxious or unpleasant stimulus is paired with an undesired behavior." This is not the definition of aversion.

b. What is the "aversion system?"

c. What is the "aversion process?"

d. Aversion is not a disorder. What is "aversion treatment?"

e. What are "mechanisms of CPA?"

f. "Mice with depression" ... mice do not have depression.

g. Refer to "nociception" in rodents rather than "pain."

3. The Introduction and the manuscript more generally should clarify:

a. Rostral versus caudal vIPAG results (e.g., TH neurons)

b. vIPAG GABA interneurons from vIPAG GABA projection neurons.

c. vIPAG and PAG seem to be used interchangeably throughout the manuscript. Be as specific as possible with each statement.

4. For each experiment, state the N for each group. N's are not clear from figure captions. In general, there is some lack of clarity regarding experimental design and timeline.

5. Why was DMSO used as the vehicle for CNO? What percent was the DMSO solution? Systemic DMSO injections are painful/aversive in rodents, especially at high concentrations. It is surprising that DMSO injections did not produce CPA on their own.

6. When was CNO administered and how many times? Prior to every pairing with "aversive" chamber? Did these details differ in the morphine WD versus the pain pinch experiments? What would one expect to be the result of pairing 100% DMSO injections with a single compartment repeatedly? Were animals injected with anything prior to pairings with the non-pain-paired compartment? Isn't it customary to do this?

7. Because the shRNA manipulations occurred 21 days prior to the start of CPA procedures, it does not necessarily show that it altered CPA learning. For example, it may have altered the pain experience itself.

8. How long after the CPA test were animals sacrificed for IHC and WB?

9. How do the authors reconcile the current data with: "Activation of TRPV1 in glutamatergic neurons in the vIPAG can alleviate pain 28."

10. Discuss the lack of changes when GABAergic neurons were activated. Would one expect higher aversion when GABA was activated? How does this agree or not agree with the literature?

11. Expand on the potentially many other possible explanations for the disparate results between this study and the Samineni paper. What were the other experimental variables that differed between the 2 studies?

12. What were the effects of AKAP150 silencing on total TRPV1? Presumably p-TRPV1 was normalized to total TRPV1?

13. This study did not inhibit AKAP150 activity as repeatedly stated or implied, it downregulated AKAP150 expression.

14. Clarify which statistical tests were used for each dataset. In the large multi-panel figures, there are at least 3 conditions/factors (virus, actuator, type of neuron transfected) if not more. It is not

clear how t-tests or 2-way ANOVAs were used for these datasets. Why were two different post hoc tests used and when were each of them used?

15. Minimize repetition from Results in the first half of Discussion section.

Minor Revisions

- Use section subtitles that summarize the main result.
- Line 19: Change the wording "drug addict" to people with substance use disorder.
- Line 51: The reference the authors provided doesn't address opposing effect on opioid in the vIPAG.
- Line 248: regarding the statement that virus did not affect the baseline compartment preference, how many days after AAV injection/transfection did testing begin and end?
- Line 326: This is the wrong reference. "The upregulation of p-TRPV1 in the vIPAG in aversion rodents was greatly dependent on AKAP150 function, which was consistent with a 6 previous study 45." It seems that the correct reference should be number 44.

Dear reviewers,

Please find a revised version of our manuscript originally entitled “**AKAP150/TRPV1 in**
**GABAergic neurons rather than glutamatergic neurons in the vPAG facilitates**
**conditioned place aversion in male mice**”, which we would like to resubmit for
publication as Article in *Communications Biology*.

Your comments were highly insightful and enabled us to improve the quality of our
manuscript. The language of the manuscript has been edited by the Nature Research
Editing Service. The certificated file is attached below.

In the following pages are our point-by-point responses to each of the comments.

Reviewer #1 (Remarks to the Author):

In manuscript COMMSBIO-23-0169-T, entitled "AKAP150/TRPV1 in GABAergic neurons rather
than glutamatergic neurons in the PAG is involved in conditioned place aversion in mice", the
authors contend that AKAP modulates TRPV1 phosphorylation in GABA neurons of the vPAG to
control conditioned place aversion. In this study, the authors utilize a well-documented model of
rodent aversion to pinch and morphine withdrawal. Furthermore, they employ viral genetics to
control AKAP and TRPV1 expression in PAG neurons via stereoscopic injection of AKAP-floxed
mice. The experiments presented here provide some sound support of the authors' hypotheses, while
other results fail to utilize correct controls to produce any conclusive evidence. Furthermore, there
is an apparent lack of synonymity between the text of the paper, figure legends, and actual figures
that has the potential to exponentially increase reader ambiguity, making the entire paper difficult
to follow. The following are more specific points that bear additional examination:

We conducted new western blotting experiments on p-TRPV1 and TRPV1 and used a correct
control for p-TRPV1 this time. We carefully checked the wording of the article to make it as clear
as possible. We also sent our article to the Springer Nature language editing center to correct our
grammar and improve our wording as concisely, correctly, and authentically as possible.

1. The investigation of TRPV1 phosphorylation is mishandled. Firstly, there is no rationale as to
why Ser502 is investigated, is there a specific kinase that AKAP is scaffolding that should control
phosphorylation at that site? Furthermore, the WB analyses of p-TRPV1 is incorrect (Fig 4 and 5),
as a protein's phosphorylation status should always be normalized to total amounts of said protein,
not beta-actin. This would ensure that changes in reported phosphorylation are not due to changes
in the amount of protein being phosphorylated.

Several phosphorylation sites on TRPV1 have been identified. We chose Ser502 mainly for the
following reasons. First, previous studies^{1,2} related to the investigation of TRPV1 phosphorylation
mostly focus on the site Ser502, and the available antibodies of TRPV1 phosphorylation are mainly
antibodies of Ser502 phosphorylation. Second, CaMKII, PKC and PKA are well-characterized
protein kinases that can phosphorylate TRPV1. Of the identified TRPV1 phosphorylation sites,
Ser502 is unique because it can be phosphorylated by all three kinases. AKAP150 has been
demonstrated to mediate the anchoring of PKA and protein kinase C (PKC) to TRPV1. Thus, we
assume that Ser502 is an important target involved in the process of pTRPV1-mediated CPA. We
have acknowledged that there is no evidence to prove that phosphorylation sites other than Ser502
do not participate in the process by which pTRPV1 mediates CPA. This is a weakness in the design
of our study. We have added some discussion about this limitation of our study in the discussion
section (line 451-454).

2. The main point of this submission, and the name of the tile, implies that AKAP/TRPV1 together
modulate the behavior phenomenon, while the data only implies that they both do so, but separately.
There is no experiment performed in this project to underscore any premise that AKAP is
scaffolding anything to TRPV12 to control the behavior. There is only a collection of individual
studies that show either AKAP or TRPV1 have modulatory roles, not that these roles are in any way
connected.

It is a limitation that there are no specific experiments on the interaction between AKAP150 and
TRPV1 or how AKAP150 and TRPV1 mediate the CPA process as a whole within our study. The
research on the interaction between AKAP150 and TRPV1 in our study is mostly derived from the
literature. We simply verified that the level of TRPV1 phosphorylation was altered after
knockdown of AKAP150 expression, which is indirect evidence of the interaction between
AKAP150 and TRPV1. To be honest, it is beyond our ability to carry out experiments to confirm
how AKAP150 and TRPV1 modulate the CPA process together. We have further discussed this
limitation in the discussion section (line 455-457) and tried our best to present our thoughts
precisely. Hopefully, you will approve of our response.

3. In Fig 1, the authors require an IHC control for panels N and P. Perhaps NeuN as used elsewhere
in the submission.

According to the reviewer's comment, we have adjusted the overall content of the figures in the
manuscript. We have added IHC controls for NeuN in Fig. 2.

4. Controls for AKAP shRNA efficacy should be alluded to earlier in the results section
corresponding to Fig 3. The authors invite reader ambiguity by not plainly identifying supporting
data that is necessary to validate their animal model.

We have adjusted the overall content of the figures in the manuscript. We have added the results of
the western blotting assay to demonstrate the efficacy of AKAP150-shRNA in Fig. 4 along with the
results of IHC and behavior tests.

5. In the results section corresponding to Fig 5, the authors state that AKAP knockdown can reduce
the expression of p-TRPV1. Do the authors mean for pSer502? pSer800? Perhaps pSer116? Maybe
one of the phosphorylatable Tyr residues of TRPV1. This invites more ambiguity by the reader to
determine what the authors conclude.

It is our mistake that the statement about Fig. 5 is not precise and clear enough. We have simply
verified the expression level of pSer502-TRPV1, and we are not sure about the role of other
phosphorylation sites in our study. We have altered the statement to "pSer502-TRPV1 decreased
significantly after silencing AKAP150 in GABAergic neurons of the vIPAG" (line 335-336).

Reviewer #2 (Remarks to the Author):

1. It is difficult to determine from the data or the discussion whether the authors believe that it is the
synaptic input to vIPAG neurons that is important for determining CPA, or whether it is the effect
of their manipulations on the glutamate or GABA neurons themselves within the vIPAG that is
important. Can the chemogenetic manipulations discriminate altered glutamatergic and GABAergic
neuron activity versus effects on synaptic terminal release of these neurotransmitters? Specifically,
can the authors state whether these manipulations targeted neurotransmitter release in the vIPAG,
or just the somatic activity of GABAergic versus glutamatergic neurons in the vIPAG? As there is
no direct physiological readout of the activity of these neurons or of synaptic transmission, it would
be useful to either provide this, or to address this issue in some other way, such as commenting on
it in the discussion.

Chemogenetic manipulations are comprised of a transgenic actuator for a cellular pathway that is
targeted to specific cell populations and can be rapidly switched on or off by delivery of a chemical
ligand. We can only utilize the specificity of the viral promoter to achieve cell population selection,
and that is the way we selectively interfere with glutamatergic or GABAergic neurons to the
greatest extent. Chemogenetic manipulations have been widely used in the modulation of specific
neuron activity, although we cannot exclude the possibility of virus overflow. Since
electrophysiology data were not provided in our study, we are not sure whether the manipulation
of AKAP150 expression affects synaptic transmission.

Since we did not include the patch clamp in the experimental design, it is difficult for us to
determine whether the impact on CPA responses is due to altered glutamatergic and GABAergic
neuron activity or the effects on synaptic terminal release of neurotransmitters. This is a limitation
of our study, and more experiments are needed for further exploration.

We have only explored the effect of AKP150 and pSer502-TRPV1 on the CPA response. We have
not excluded the possibility that AKAP150 might affect CPA through other mechanisms, such as
calcineurin, PKA, and PKC. This is a limitation of our study, and more experiments are needed for
further exploration. We have included comments on it in the discussion section (line 458-463).

2. Figures 2-3 are unnecessarily complex because they show all of the pre-paired and post-paired
data, rather than just providing a single example of this followed by the simplified aversion score. I
think that it is sufficient to show the pre- and post-paired data once as an example, and then provide
the aversion score data in the rest of the figures.

Thank you for the advice. We have adjusted the figures in the manuscript to make them more
effective.

3. line 66. “this finding indicated that different CPA model mechanisms are not the same, and the
role of PAG in CPA is complex and complicated.” As the terms “complex” and “complicated” mean
the same thing in this context, this sentence should be rewritten for clarity.

We have rewritten this sentence (line 73-75).

4. line 250. “after 21 days, when AAV transgenic expression was stable,”. Do the authors mean to
say, “when the expression of gene expression was stable”? This should be corrected.

Generally, AAV infection and stable expression take up to three weeks in mice. We have changed
the formulation (line 201).

Reviewer #3 (Remarks to the Author):

1. In the title:

a. State “in male mice.”

We have amended it (line 2).

b. “is involved in” is not informative... state direction of the result.

We have changed the title to “AKAP150/TRPV1 in GABAergic neurons rather than glutamatergic
neurons in the PAG facilitates conditioned place aversion in male mice” (line 1-3).

2. The manuscript is extremely hard to follow, and there many unclear points and details. Please
carefully re-write, perhaps with the assistance of a native English speaker. For example, in the
Abstract and in the manuscript more generally, there are some sweeping statements and definitions
that are inaccurate, and also loose swapping of terminology, lack of clarity, etc., for example:

a. The Abstract defines aversion as “the process by which a noxious or unpleasant stimulus is paired
with an undesired behavior.” This is not the definition of aversion.

We have changed the wording to “Aversion refers to a feeling of strong dislike or avoidance toward
a particular stimulus or situation” (line 18-19).

- b. What is the “aversion system?”
- We have changed “aversion system” to “aversive behavior”.
- c. What is the “aversion process?”
- We have changed “aversion process” to “aversive behavior”.
- 140 d. Aversion is not a disorder. What is “aversion treatment?”
- We have amended it to the “CPA response”.
- e. What are “mechanisms of CPA?”
- We have amended it to “The effect of PAG in CPA is not fully understood” (line 62).
- f. “Mice with depression” ... mice do not have depression.
- We have changed the wording to “the murine models of cold stress-induced nociception and
depression” to match the original reference (line 85).
- 147 g. Refer to “nociception” in rodents rather than “pain.”
- We have amended “pain modulation” to “nociception modulation”. “Itch and pain” was changed
to “Itch and nociception”
- 3. The Introduction and the manuscript more generally should clarify:
- a. Rostral versus caudal vIPAG results (e.g., TH neurons)
- The test part of the PAG in the experiment was the caudal vIPAG part. We added this information
to the Methods section (line 173).
- b. vIPAG GABA interneurons from vIPAG GABA projection neurons.
- The GABA neuron promoter we used is *vgat1*, which is generally used to identify GABA
interneurons but not projection neurons in the vIPAG brain area. We added this information to
the Methods section (line 178-179).
- c. vIPAG and PAG seem to be used interchangeably throughout the manuscript. Be as specific as
possible with each statement.
- Thank you for pointing this out. Based on the results of published studies, the specific site that we
aim to study is the ventrolateral periaqueductal gray (vIPAG). We have gone through the whole
manuscript and made sure all the words related to our study design stick to vIPAG. “PAG”s used in
the introduction and discussion section are due to respect for the original study that we referenced.
- 4. For each experiment, state the N for each group. N's are not clear from figure captions. In general,
there is some lack of clarity regarding experimental design and timeline.

We have added the N for each group in the figure legends, as well as some essential experimental
design and timeline information.

5. Why was DMSO used as the vehicle for CNO? What percent was the DMSO solution? Systemic
DMSO injections are painful/aversive in rodents, especially at high concentrations. It is surprising
that DMSO injections did not produce CPA on their own.

We did not provide a clear description, which led to misunderstandings. The dissolution of CNO
followed the methods in this reference with minor modifications³. First, we dissolved CNO in 0.5%
DMSO and then diluted CNO to the appropriate ratio with physiological saline. Finally, the
concentration of DMSO was 0.05%. DMSO (0.05%) was used as a control for the vehicle. Injecting
such a concentration into the abdominal cavity of mice would not cause pain or aversion in rodents.
To avoid potential misunderstandings, we have changed DMSO to vehicle in the figures and added
detailed descriptions in the methodology to ensure that readers can understand the details. We
added the necessary information in the Methods section (line 184-185).

6. When was CNO administered and how many times? Prior to every pairing with "aversive"
chamber? Did these details differ in the morphine WD versus the pain pinch experiments? What
would one expect to be the result of pairing 100% DMSO injections with a single compartment
repeatedly? Were animals injected with anything prior to pairings with the non-pain-paired
compartment? Isn't it customary to do this?

Because the solvent of CNO contains a small amount of DMSO (0.05% DMSO, as mentioned in the
previous comments), to ensure the rigor of the control group, the control group was injected with
0.05% DMSO intraperitoneally. When the experimental group was injected with CNO, the control
group was injected with 0.05% DMSO. The number of injections for both groups was the same.
We have added a detailed description of this section in the methodology (line 184-185).

7. Because the shRNA manipulations occurred 21 days prior to the start of CPA procedures, it does
not necessarily show that it altered CPA learning. For example, it may have altered the pain
experience itself.

After the shRNA manipulations, we tested the basic preference again before the establishment of
the CPA model. We confirmed that the basic preference was not changed through comparison of
the data before and after shRNA manipulation. We show these data in the supplementary material
(Fig. S1). We also tested that the basic pain threshold of mice after knocking down AKAP150 did
not change. As this is not the focus of this study, we did not include a detailed description or data
in the manuscript. We added this expression in the article (line 203).

The experimental protocols and results of the mechanical allodynia and thermal hyperalgesia
assays are shown as follows.

1. Mechanical allodynia Mechanical sensitivity was assessed using von Frey hairs (0.04, 0.16,
0.4, 0.6, 1.0, 2.0, 4.0 g) with an updown method. The 0.6 g stimulus was applied first. If the
mouse did not retract or move its hind paw following the stimulus, the next stronger stimulus
was applied. If the hind paw responded negatively, a weaker stimulus was applied. Each
stimulus consisted of a 2-3 s application of the Von Frey hair to the lateral surface of the foot
with a 5-min interval between stimuli. Quick withdrawal or licking of the hindpaw in response

to the stimulus was considered a positive response.
 2. Hot plate analgesia assay Analgesia was measured using a 52 °C hot plate apparatus (UGO
 Basile). Mice were habituated in the room for 1 h. For the hot plate analgesia study, mice were
 placed on the hot plate, and the latencies to jump or lick the hind paw were recorded. A cutoff
 time of 45 s was used to avoid tissue damage and inflammation.

8. How long after the CPA test were animals sacrificed for IHC and WB?

Two hours after the last match of the pinch pain model (Fig. 1b Day 6) or morphine withdrawal
 model (Fig. 1c Day 7), the mice were anesthetized and euthanized for WB and IHC brain vIPAG
 specimen harvest. We added this information to the methodology (line 230-232).

9. How do the authors reconcile the current data with: “Activation of TRPV1 in glutamatergic
 neurons in the vIPAG can alleviate pain 28.”

Starowicz reported that a TRPV1 antagonist in the vIPAG facilitates nociceptive responses, while a
 TRPV1 agonist in the vIPAG promotes antinociception. This seems to be inconsistent with our study,
 since we state that only silencing TRPV1 in GABAergic neurons of the vIPAG can attenuate the CPA
 response induced by pinch pain or morphine withdrawal. However, there are differences between
 these two studies in experimental design. First, Starowicz’s study focused on the modulatory effect
 of TRPV1 in response to pain stimuli, while our study focused on the aversion response induced by
 pain. The aversion response induced by pain is not equal to the pain stimulus, meaning that the
 aversion and the pain stimuli may both relate to the activity of the vIPAG but may not receive
 modulation from the same region in the vIPAG. Second, Starowicz’s study mainly investigated the
 instant effect of nociception or antinociception after injecting the TRPV1 antagonist or agonist,
 while our study selectively knocked down the expression of TRPV1 by virus overexpression with
 the GABA neuron promoter Vgat1. Neither TRPV1 agonist nor antagonist is selective toward TRPV1

receptors, which is completely different from the selective knockdown by shRNA. There are
possibilities that TRPV1 mediates the CPA response through other pathways, which requires
further experiments to clarify. Therefore, we still have a long way to go to reveal the whole picture.
We have added more content discussing the discrepancy between other studies and our study in
the discussion section to present our perspective (line 423-438).

10. Discuss the lack of changes when GABAergic neurons were activated. Would one expect
higher aversion when GABA was activated? How does this agree or not agree with the literature?

From the literature review, we found that activation of GABAergic neurons facilitates nociception.
We demonstrated that activation of GABAergic neurons cannot affect aversion induced by
hindpaw pinch pain. Since acute pain stimuli cannot induce aversive behavior, we hypothesize that
the activation of GABAergic neurons has an impact on the modulation of immediate stimuli such
as pain rather than on the modulation of aversion. More experiments are needed for verification.

11. Expand on the potentially many other possible explanations for the disparate results between
this study and the Samineni paper. What were the other experimental variables that differed between
the 2 studies?

In reference 7, they found that selective modulation of GABAergic or glutamatergic neurons
demonstrates an inverse regulation of nociceptive behaviors by these cell populations. Selective
chemogenetic activation of glutamatergic neurons or inhibition of GABAergic neurons in the vIPAG
suppresses nociception. In contrast, inhibition of glutamatergic neurons or activation of GABAergic
neurons in the vIPAG facilitates nociception. In our study, we demonstrate that inhibition of vIPAG
GABAergic neurons, but not glutamatergic neurons, significantly attenuated the conditioned place
aversion induced by hindpaw pain pinch or naloxone-precipitated morphine withdrawal.
Samineni's study focused on the role of GABAergic or glutamatergic neurons from the vIPAG in
acute pain modulation. However, our study focused on the aversion modulation induced by
repetitive pinch pain or morphine withdrawal. Acute pain stimuli cannot necessarily induce
aversion. We assume that activation of GABAergic or glutamatergic neurons or inhibition of
glutamatergic neurons interferes with acute pain modulation but not aversion induced by hindpaw
pain or morphine withdrawal.

In reference 13, they found that the activity of GABAergic and glutamatergic neurons in the PAG
is modulated in an opposing manner during chloroquine-evoked scratching. Activation of PAG
GABAergic neurons or inhibition of glutamatergic neurons resulted in attenuation of scratching in
both acute and chronic pruritis. This is not completely consistent with our results. Itch and
nociceptive pain exhibit an inverse and reciprocal relationship at the behavioral level. Although
itch and nociceptive pain and itch are processed and transmitted by similar neuroanatomical
substrates at peripheral, spinal and supraspinal sites, they are discriminated and processed
distinctly at the cellular level. This may be the possible explanation for the significant difference
between the two results.

12. What were the effects of AKAP150 silencing on total TRPV1? Presumably p-TRPV1 was
normalized to total TRPV1?

Silencing AKAP150 does not have an impact on the expression of total TRPV1. We have performed
all the western blotting assays again and added the data for total TRPV1, to which p-TRPV1 is
normalized.

13. This study did not inhibit AKAP150 activity as repeatedly stated or implied, it downregulated
AKAP150 expression.

AKAP150 can regulate the activity of downstream kinases, such as PKA, PKC and calcineurin. We
have not found methods to inhibit the activity of AKAP150. We can only inhibit AKAP150 function
by downregulating its expression.

14. Clarify which statistical tests were used for each dataset. In the large multi-panel figures, there
are at least 3 conditions/factors (virus, actuator, type of neuron transfected) if not more. It is not
clear how t-tests or 2-way ANOVAs were used for these datasets. Why were two different post hoc
tests used and when were each of them used?

We have added detailed descriptions in the statistics section (line 235-239), and we have also
added necessary descriptions in the figure legends to ensure a smooth reading of the article. The
comparison of behavior and WB between two groups was conducted using t tests. The comparison
of mean velocity was conducted using two-way repeated-measures ANOVA followed by
Bonferroni's post hoc test. The comparison of aversion scores among multiple groups was
conducted using one-way ANOVA followed by Bonferroni's post hoc test. We have deleted
Dunnett's T3 post hoc test.

15. Minimize repetition from Results in the first half of Discussion section.

We have deleted the repetitive part in the discussion.

Minor Revisions

- Use section subtitles that summarize the main result.

Thank you for the advice. We have adjusted section subtitles in the results section.

- Line 19: Change the wording "drug addict" to people with substance use disorder.

We have amended it.

- Line 51: The reference the authors provided doesn't address opposing effect on opioid in the
vIPAG.

We have rewritten these sentences to make it clearer and more precise as follow. Evidence indicates
that subpopulations of the ventrolateral periaqueductal gray (vIPAG) exert different effects on
nociception⁴. We deleted the other two references here (line 55-56).

- Line 248: regarding the statement that virus did not affect the baseline compartment preference,
how many days after AAV injection/transfection did testing begin and end?

The stable expression of AAV takes 3 weeks, and the baseline CPP determination of all mice is
performed between 21 and 22 days after AAV injection. We have added the necessary information
in the methodology (line 201-202).

- Line 326: This is the wrong reference. “The upregulation of p-TRPV1 in the vIPAG in aversion
rodents was greatly dependent on AKAP150 function, which was consistent with a previous study
45.” It seems that the correct reference should be number 44.

We have amended the reference to number 43⁵ and made the wording more precise (line 337-
339).

Revisions in the manuscript are shown in red front. We hope that the revision in the
manuscript and our accompany responses will be sufficient to make our manuscript
suitable for publication in *Communications Biology*.

We shall look forward to hearing from you at your earliest convenience.

Yours sincerely,

Handong Ouyang

Address: 651 Dongfeng Road East, Guangzhou, China.

Tel: 86-20-87343060.

E-mail: ouyhd@sysucc.org.cn

Reference

1. Z. Guo, et al. Spinophilin negatively controlled the function of transient receptor potential vanilloid
1 in dorsal root ganglia neurons of mice. *Eur J Pharmacol* **863**.172700-172700 (2019).

2. D. L. Liu, W. T. Wang, J. L. Xing, S. J. Hu. Research progress in transient receptor potential vanilloid
1 of sensory nervous system. *Neurosci Bull* **25**.221-227 (2009).

3. L. A. DeNardo, et al. Temporal evolution of cortical ensembles promoting remote memory retrieval.
*Nat Neurosci* **22**.460-469 (2019).

4. V. K. Samineni, et al. Divergent Modulation of Nociception by Glutamatergic and GABAergic
Neuronal Subpopulations in the Periaqueductal Gray. *eNeuro* **4**.(2017).

5. N. A. Jeske, et al. A-kinase anchoring protein mediates TRPV1 thermal hyperalgesia through PKA
phosphorylation of TRPV1. *Pain* **138**.604-616 (2008).

Reviewers' comments:

Reviewer #1 (Remarks to the Author):

The authors remedied some of the ambiguities in this resubmission, but simply inserted a discussion paragraph to explain away others that must be addressed to fix the main problem(s) with this manuscript.

1. Lack of a rationale for Ser502 as the TRPV1 phosphorylation site for investigation. The authors cannot simply state that they investigated this site because it is the most researched. What does it have to do with the physiology being examined? Have other reports identified this site to be important in AKAP-scaffolded post-translational modifications in this part of the neural system?

2. Lack of data supporting a scaffolding role for TRPV1 in this model. The data supports that AKAP has a role, and that TRPV1 has a role in some of the experiments, but there is no data that supports AKAP scaffolds TRPV1 in this phenomenon. This key work is necessary, and was ignored by the authors.

Reviewer #2 (Remarks to the Author):

In this report the authors study the involvement of TRPV1 receptors, the anchoring protein AKAP150, and GABAergic versus glutamatergic inputs to the ventrolateral periaqueductal gray (vlPAG) in 2 distinct conditioned place aversion (CPA) models. The first model examines CPA caused by hind paw pinch, and the second by naloxone-precipitated opiate withdrawal. The rationale for examining AKAP150 is derived from observations that it can interact with signaling enzymes such as the regulatory subunit of protein kinase A (PKA) and PKC to anchor these molecules to the membrane and to aid phosphorylation of TRPV1Rs, which are ubiquitously implicated in nociception. Using a variety of biochemical and genetic tools, including chemogenetics, and protein knockdown, the authors report that selective inhibition of vlPAG GABAergic neurons reduced CPA in both models, whereas inhibition of vlPAG glutamatergic neurons had no effect on the pain or morphine withdrawal CPA responses. The authors also show that AKAP150 protein expression levels and phosphorylation of TRPV1 were increased in both CPA models, and that selective knockdown of AKAP150 with shRNA in GABAergic neurons also decreased both forms of CPA. In general, this is a thorough and rigorous study that provides evidence for the involvement of AKAP150 in the phosphorylation of TRPV1 channels in GABAergic neurons of the vlPAG. This suggests that this pathway may be a candidate for consideration as a target for alleviating conditioned aversion.

Comments.

In my prior review of this manuscript, I asked whether physiological endpoints had been considered for manipulation of GABAergic vs glutamatergic circuitry. Whereas the authors understandably did not provide these data in the revision they did indicate in the revised discussion that this is a limitation of the study and suggest that it might be important to determine in subsequent research. In general, I find that the addition of this paragraph describing the limitations of this study encapsulates the primary concerns I had in this regard. The authors have also responded by revising figures 2-3, that were more complex than necessary, rendering the data more accessible.

In reading through the revised manuscript, it seems as though the authors have been very responsive to most of the concerns raised in the initial review. Because of this I have no recommendations for further revision.

Reviewer #3 (Remarks to the Author):

In this revised manuscript, the authors addressed most of my previous concerns, but some concerns remain unaddressed in the manuscript:

1. In the manuscript, discuss the lack of changes when GABAergic neurons were activated. Would one expect higher aversion when GABAergic neurons are activated? How does this agree or not agree with literature?
2. In the manuscript, expand on the potentially many other possible explanations for the disparate results between this study and the Samineni paper. What were the other experimental variables that differed between the 2 studies?
3. In the manuscript, clarify which statistical tests were used for each dataset. In the large multi-panel figures, there are at least 3 conditions/factors (virus, actuator, type of neuron transfected) if not more. It is not clear how t-tests or 2-way ANOVAs were used for these datasets. Why were two different post-hoc tests used and when were each of them used?
4. Use section subtitles that summarize the main result of each sub-section.
5. Line 61: Instead of "effect" state "the role of the vIPAG in CPA is not fully understood."

Dear reviewers,

Please find a revised version of our manuscript originally entitled “AKAP150/TRPV1 in GABAergic neurons rather than glutamatergic neurons in the vIPAG facilitates conditioned place aversion in male mice”, which we would like to resubmit for publication as Article in *Communications Biology*.

Your comments were highly insightful and enabled us to improve the quality of our manuscript. In the following pages are our point-by-point responses to each of the comments.

Reviewer #1:

The authors remedied some of the ambiguities in this resubmission, but simply inserted a discussion paragraph to explain away others that must be addressed to fix the main problem(s) with this manuscript.

1. Lack of a rationale for Ser502 as the TRPV1 phosphorylation site for investigation. The authors cannot simply state that they investigated this site because it is the most researched. What does it have to do with the physiology being examined? Have other reports identified this site to be important in AKAP-scaffolded post-translational modifications in this part of the neural system?

We chose Ser502 as the TRPV1 phosphorylation site for investigation not only because it is the most studied but because it is an important site taking part in the interaction between AKAP150 and phosphorylated TRPV1. AKAP150 has been demonstrated to mediate the anchoring of PKA and PKC to TRPV1. CaMKII, PKC and PKA are well-characterized protein kinases that can phosphorylate TRPV1. Of the identified TRPV1 phosphorylation sites, Ser502 is unique because it can be phosphorylated by all three kinases¹. Zhang et al. reported that AKAP79/150-mediated TRPV1 trafficking depends on Ser502 phosphorylation². Thus, we picked the site Ser502.

2. Lack of data supporting a scaffolding role for TRPV1 in this model. The data supports that AKAP has a role, and that TRPV1 has a role in some of the experiments, but there is no data that supports AKAP scaffolds TRPV1 in this phenomenon. This key work is necessary, and was ignored by the authors.

Previous studies reported that AKAP150 could mediate trafficking of PKA and PKC to TRPV1, which is required for phosphorylation of the channel^{3, 4}, and the trafficking of PKA to TRPV1 by AKAP150 may be critical in the development of hyperalgesia⁵. Based on these studies, we investigated the relationship between the expression of AKAP150 and TRPV1 phosphorylation in the CPA models studied in our research. In our study, we demonstrate that AKAP150 is upregulated in the chosen CPA models and knockdown of AKAP150 attenuates the CPA response. Silencing TRPV1 in GABAergic neurons of the vIPAG attenuates the CPA response and phosphorylated TRPV1 expression is downregulated after silencing AKAP150. Thus, we assume that AKAP150/p-TRPV1 in GABAergic neurons of the vIPAG could be a pathway for developing strategies for attenuating the CPA response. In our study, we are not trying to testify the anchoring reaction between AKAP150 and TRPV1. We have tried our best to prove that AKAP150 and TRPV1 in

vIPAG GABAergic neurons affect aversion, and expression level of AKAP150 interferes with the TRPV1 phosphorylation. We do understand that the direct data proving AKAP150 scaffolds TRPV1 would provide more solid evidence to support our statement. We would try our best to improve our ability and solve the problem in the future. We hope that you would accept our efforts and results.

Reviewer #2:

In this report the authors study the involvement of TRPV1 receptors, the anchoring protein AKAP150, and GABAergic versus glutamatergic inputs to the ventrolateral periaqueductal gray (vIPAG) in 2 distinct conditioned place aversion (CPA) models. The first model examines CPA caused by hind paw pinch, and the second by naloxone-precipitated opiate withdrawal. The rationale for examining AKAP150 is derived from observations that it can interact with signaling enzymes such as the regulatory subunit of protein kinase A (PKA) and PKC to anchor these molecules to the membrane and to aid phosphorylation of TRPV1Rs, which are ubiquitously implicated in nociception. Using a variety of biochemical and genetic tools, including chemogenetics, and protein knockdown, the authors report that selective inhibition of vIPAG GABAergic neurons reduced CPA in both models, whereas inhibition of vIPAG glutamatergic neurons had no effect on the pain or morphine withdrawal CPA responses. The authors also show that AKAP150 protein expression levels and phosphorylation of TRPV1 were increased in both CPA models, and that selective knockdown of AKAP150 with shRNA in GABAergic neurons also decreased both forms of CPA. In general, this is a thorough and rigorous study that provides evidence for the involvement of AKAP150 in the phosphorylation of TRPV1 channels in GABAergic neurons of the vIPAG. This suggests that this pathway may be a candidate for consideration as a target for alleviating conditioned aversion.

Comments.

In my prior review of this manuscript, I asked whether physiological endpoints had been considered for manipulation of GABAergic vs glutamatergic circuitry. Whereas the authors understandably did not provide these data in the revision they did indicate in the revised discussion that this is a limitation of the study and suggest that it might be important to determine in subsequent research. In general, I find that the addition of this paragraph describing the limitations of this study encapsulates the primary concerns I had in this regard. The authors have also responded by revising figures 2-3, that were more complex than necessary, rendering the data more accessible.

In reading through the revised manuscript, it seems as though the authors have been very responsive to most of the concerns raised in the initial review. Because of this I have no recommendations for further revision.

Reviewer #3:

In this revised manuscript, the authors addressed most of my previous concerns, but some concerns remain unaddressed in the manuscript:

1. In the manuscript, discuss the lack of changes when GABAergic neurons were activated. Would one expect higher aversion when GABAergic neurons are activated? How does this agree or not agree with literature?

In most cases, activation and inhibition of the same neurons are more likely to exert opposing effects. However, activation and inhibition of the same neurons might exert effects in the same direction or exert no effect in some cases. For instance, Samineni⁶ found that activating GABAergic or inhibiting glutamatergic neurons in vIPAG causes thermal hypersensitivity and inhibiting GABAergic or activating glutamatergic neurons attenuates thermal sensitivity using cell type-specific chemogenetic manipulations. However, they also demonstrated that activating GABAergic or inhibiting glutamatergic neurons in vIPAG causes mechanical hypersensitivity, while inhibiting GABAergic or activating glutamatergic neurons has no effect on mechanical sensitivity. Though previous study demonstrates that activation of vIPAG GABAergic neurons attenuates place aversion in the itch CPA model⁷. In our study, inhibition of GABAergic neurons attenuates the CPA response, while activation of GABAergic neurons cannot affect or alter the CPA response induced by hindpaw pain pinch or naloxone-precipitated morphine withdrawal. We believe that the different CPA models studied in the two studies should be the main reason for the discrepancy between the results and the result of our study is logical. We have expanded the content in the discussion section (line 375-378).

2. In the manuscript, expand on the potentially many other possible explanations for the disparate results between this study and the Samineni paper. What were the other experimental variables that differed between the 2 studies?

There are two Samineni papers mentioned in the manuscript. The discussions about the disparate results between our study and the Samineni papers are shown in the discussion section (line 370-395).

In reference 7, they found that selective modulation of GABAergic or glutamatergic neurons demonstrates an inverse regulation of nociceptive behaviors by these cell populations. Selective chemogenetic activation of glutamatergic neurons or inhibition of GABAergic neurons in the vIPAG suppresses nociception. In contrast, inhibition of glutamatergic neurons or activation of GABAergic neurons in the vIPAG facilitates nociception. In our study, we demonstrate that inhibition of vIPAG GABAergic neurons, but not glutamatergic neurons, significantly attenuated the conditioned place aversion induced by hindpaw pain pinch or naloxone-precipitated morphine withdrawal. Samineni's study focused on the role of GABAergic or glutamatergic neurons from the vIPAG in acute pain modulation. However, our study focused on the aversion modulation induced by repetitive pinch pain or morphine withdrawal. Acute pain stimulus is not necessarily equivalence to aversion. We assume that activation of GABAergic or glutamatergic neurons or inhibition of glutamatergic neurons interferes with acute pain modulation but not aversion induced by hindpaw pain or morphine withdrawal.

In reference 13, they found that the activity of GABAergic and glutamatergic neurons in the PAG is modulated in an opposing manner during chloroquine-evoked scratching. Activation of PAG GABAergic neurons or inhibition of glutamatergic neurons resulted in attenuation of scratching in both acute and chronic pruritis. This is not completely consistent with our results. Itch and nociceptive pain exhibit an inverse and reciprocal relationship at the behavioral level. Although itch and nociception pain and itch are processed and transmitted by similar neuroanatomical substrates at peripheral, spinal and supraspinal sites, they are discriminated and processed distinctly at the cellular level. This may be the possible explanation for the significant difference between the two

results.

3. In the manuscript, clarify which statistical tests were used for each dataset. In the large multi-panel figures, there are at least 3 conditions/factors (virus, actuator, type of neuron transfected) if not more. It is not clear how t-tests or 2-way ANOVAs were used for these datasets. Why were two different post-hoc tests used and when were each of them used?

The statistical tests used for each dataset are pointed out in the figure legends. Comparison of the aversion score or the expression level of western blotting assay between two groups is conducted by two sample t-test. Comparison of the aversion score among three or more than three groups is conducted by one-way ANOVA. Comparison of the mean velocity is conducted by two-way ANOVA because each comparison involves two factors.

4. Use section subtitles that summarize the main result of each sub-section.

We have made changes in the section subtitles to summarize the main result of each sub-section.

5. Line 61: Instead of “effect” state “the role of the vIPAG in CPA is not fully understood.”

We have amended it (line 61).

Revisions in the manuscript are shown in red front. We hope that the revision in the manuscript and our accompany responses will be sufficient to make our manuscript suitable for publication in *Communications Biology*.

We shall look forward to hearing from you at your earliest convenience.

Yours sincerely,

Handong Ouyang

Address: 651 Dongfeng Road East, Guangzhou, China.

Tel: 86-20-87343060.

E-mail: ouyhd@sysucc.org.cn

Reference

1. Z. Guo, et al. Spinophilin negatively controlled the function of transient receptor potential vanilloid 1 in dorsal root ganglia neurons of mice. *Eur J Pharmacol* **863**.172700 (2019).
2. X. Zhang, L. Li, P. A. McNaughton. Proinflammatory Mediators Modulate the Heat-Activated Ion Channel TRPV1 via the Scaffolding Protein AKAP79/150. *Neuron* **59**.450-461 (2008).
3. K. E. Brandao, M. L. Dell'Acqua, S. R. Levinson. A-kinase anchoring protein 150 expression in a specific subset of TRPV1- and CaV 1.2-positive nociceptive rat dorsal root ganglion neurons. *J Comp Neurol* **520**.81-99 (2012).
4. M. K. Chung, J. Lee, J. Joseph, J. Saloman, J. Y. Ro. Peripheral group I metabotropic glutamate

receptor activation leads to muscle mechanical hyperalgesia through TRPV1 phosphorylation in the rat. *J Pain* **16**:67-76 (2015).

5. R. Efendiev, A. Bavencoffe, H. Z. Hu, M. X. Zhu, C. W. Dessauer. Scaffolding by A-kinase anchoring protein enhances functional coupling between adenylyl cyclase and TRPV1 channel. *J Biol Chem* **288**:3929-3937 (2013).

6. V. K. Samineni, et al. Divergent Modulation of Nociception by Glutamatergic and GABAergic Neuronal Subpopulations in the Periaqueductal Gray. *eNeuro* **4**.(2017).

7. V. K. Samineni, J. G. Grajales-Reyes, S. S. Sundaram, J. J. Yoo, R. W. t. Gereau. Cell type-specific modulation of sensory and affective components of itch in the periaqueductal gray. *Nat Commun* **10**:4356 (2019).

REVIEWERS' COMMENTS:

Reviewer #3 (Remarks to the Author):

All of my prior comments have been addressed. I have no further comments.